# OTTERS: AN ENERGY-EFFICIENT SPIKING TRANSFORMER VIA OPTICAL TIME-TO-FIRST-SPIKE ENCODING

**Zhanglu Yan**[1,*,†], **Jiayi Mao**[2,*], **Qianhui Liu**[3], **Fanfan Li**[2], **Gang Pan**[4], **Tao Luo**[5], **Bowen Zhu**[2], **Weng-Fai Wong**[1]

[1]Department of Computer Science, National University of Singapore,
[2]School of Engineering, Westlake University,
[3]School of Artificial Intelligence, Shandong University,
[4]College of Computer Science and Technology, Zhejiang University,
[5]Agency for Science, Research and Technology, Singapore.

## ABSTRACT

Spiking neural networks (SNNs) promise high energy efficiency, particularly with time-to-first-spike (TTFS) encoding, which maximizes sparsity by emitting at most one spike per neuron. However, this energy advantage is often unrealized because inference requires evaluating a temporal decay function and then multiplying the result by the synaptic weights. This paper challenges this costly approach by repurposing a physical hardware 'bug', namely, the natural signal decay in optoelectronic devices, as the core computation of TTFS. We fabricated a custom indium oxide optoelectronic synapse that demonstrates how its intrinsic physical decay directly implements the required temporal function. By treating the device's analog output as the fused product of the synaptic weight and temporal decay, optoelectronic synaptic TTFS (named Otters) eliminates these expensive digital operations. To use the Otters' paradigm in complex architectures such as the transformer, which are challenging to train directly due to sparsity, we introduce a novel quantized neural network-to-SNN conversion algorithm. This complete hardware-software co-design enables our model to achieve state-of-the-art accuracy across seven GLUE benchmark datasets and demonstrates a $1.77\times$ improvement in energy efficiency over previous leading SNNs, based on a comprehensive analysis of compute, data movement, and memory access costs using energy measurements from a commercial 22nm process. Our work thus establishes a new paradigm for energy-efficient SNNs that translates fundamental device physics directly into powerful computational primitives. All codes and data are open source[1].

## 1 INTRODUCTION

Large language models (LLMs) have demonstrated remarkable capabilities, yet their immense computational and energy costs hinder deployment on resource-constrained edge devices (Lin et al., 2023; Jegham et al., 2025). This critical challenge has spurred research on more efficient, brain-inspired architectures, with spiking neural networks (SNNs) emerging as a promising candidate (Xing et al., 2024a; Tang et al., 2025; Liu et al., 2024; 2025). SNNs are known for their potential energy efficiency, arising from sparse, event-driven computations that use addition rather than expensive multiplications. However, realizing this efficiency in practice is complex and depends heavily on the encoding scheme used (Yan et al., 2024). In conventional rate-coded SNNs, information is encoded in the number of spikes in a fixed time window. This approach necessitates multiple memory accesses for weights and frequent data movement on each spikes, which may negate the benefits of the sparse computation. Temporal coding schemes, specifically, time-to-first-spike (TTFS), offer

---

[1]https://github.com/zhangluyan9/ICLR26Otters
†Correspondence to Zhanglu Yan. Email: zlyan@nus.edu.sg, * Equal contribution.

a potentially more efficient alternative. By encoding information in the precise timing of a single spike, a TTFS-SNN neuron fires at most once per activation cycle. This maximizes sparsity, dramatically reducing spike count and the associated data movement costs, making TTFS a theoretically optimal encoding for energy efficiency (Yu et al., 2023; Zhao et al., 2025).

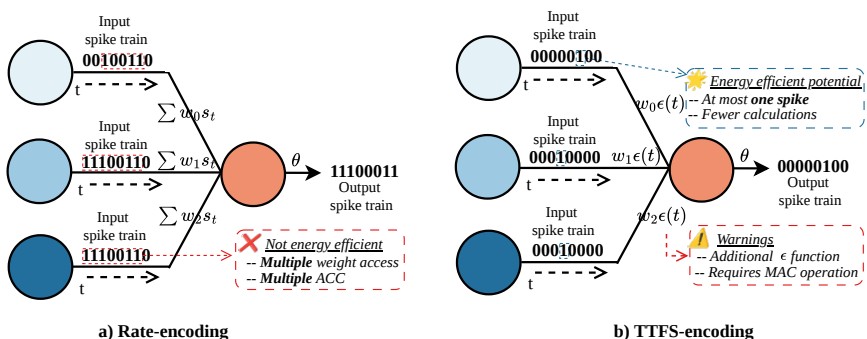

Figure 1: Rate encoding vs TTFS encoding

However, there are hidden costs behind this theoretical efficiency. In TTFS, information is encoded by mapping spike timing to importance; typically, the earlier a spike arrives, the larger the numerical value it represents. To implement this principle, the network typically requires an additional computational step to convert each spike's raw arrival time into a corresponding value. This is done using a decay function (e.g., $\epsilon(t) = e^{-t}$ or $T - t$) whose output is then multiplied by the synaptic weight $(w \cdot \epsilon(t))$ (Wei et al., 2023; Che et al., 2024b). This conversion process requires energy to compute the decay function, and it reintroduces multiplication operations that SNNs are designed to avoid. This practical drawback negates the energy savings from sparsity, thereby raising a critical question: how can we exploit the sparsity of TTFS while avoiding the costly computation? Recognizing that the costly term $w \cdot \epsilon(t)$ is a quantity that decreases predictably over time, our answer lies not in optimizing the digital computation but in finding a physical analog to simulate this process. This approach leads us to optoelectronic synapses, which are attractive for their fJ-level energy consumption and high resistance to electromagnetic interference (Li et al., 2024b). Notably, while this field has traditionally focused on suppressing their natural signal decay (volatility) to create stable memories (Alqahtani et al., 2025), we embrace this decay. We recognize it not as a bug but as the exact physical implementation of the temporal decay function required by TTFS. To implement this principle, we fabricated a custom $In_2O_3$ optoelectronic synapse. Our method, Otters, uses the natural decay of this device's optical signal to perform the required computation. This approach integrates storage and computation into a single physical step, thereby addressing the overhead of traditional TTFS.

While our Otters hardware mitigates the computational overhead of TTFS, a second major barrier remains: the inherent difficulty of training such networks, especially for complex architectures such as transformers. Directly training SNNs is challenging. In event-driven learning, error backpropagation depends on spike timing; if a neuron fires too sparsely or not at all, it fails to learn. This "over-sparsity" problem can severely limit the model's performance and even lead to training failure (Wei et al., 2023). To sidestep this issue, we employ a quantized neural network(QNN)-to-SNN conversion methodology. We first train a QNN and then convert its weights to our Otters SNN, thereby avoiding the pitfalls of direct spiking-domain training. To further maximize efficiency, we use knowledge distillation to train a highly compressed model with 1-bit weights and 1-bit key/value (KV) projections. Our complete hardware-software co-design, combining the Otters synapse with our QNN-to-SNN conversion pipeline, establishes a new state of the art for spiking language models. Evaluated on the GLUE benchmark, Otters achieves average accuracy of 3% higher than previous leading SNNs while demonstrating a 1.77-3.04× improvement in energy efficiency compared to baselines like Sorbet and SpikingLM (Tang et al., 2025; Xing et al., 2024b). Notably, this energy-efficiency gain is validated by a rigorous and comprehensive analysis that goes beyond the simplistic metrics commonly used in prior SNN research. While previous work often only counted compute operations (e.g., additions vs. multiplications), our analysis provides a more realistic estimate. It

is grounded in measurements from a commercial 22nm process and provides a full accounting of compute, data movement, and memory access costs, making our efficiency claims robust.

We also investigate the Otters paradigm's sensitivity to the hardware noise inherent in analog devices. Our initial analysis shows that, on the SST-2 benchmark, the baseline model's performance begins to degrade when key physical parameters vary by around 5%. To improve robustness, we propose Hardware-Aware Training, a method that introduces different levels of simulated Gaussian noise during training. This approach enhances the model's resilience, enabling it to maintain robust performance under noisy conditions and providing a practical path toward real-world deployment.

## 2 PRELIMINARY

### 2.1 OPTOELECTRONIC SYNAPSE

An optoelectronic synapse is a neuromorphic device that emulates biological synaptic functions by using optical signals to modulate its electrical conductance. These devices are renowned for their potential for extreme energy efficiency, broader bandwidth and faster signal transmission in neuromorphic computing, which are key advantages over purely electronic counterparts (Xie et al., 2024; Wang et al., 2023). Recent studies have reported energy consumption reaching the femtojoule (fJ)/spike level, comparable to that of biological synapses and substantially lower than that of conventional CMOS neuron devices (Shi et al., 2022; Wang et al., 2024). Among various implementations, oxide thin-film transistors (TFT) are regarded as viable candidates for optoelectronic synapses due to their low leakage current and capability for large-area, flexible fabrication. Solution-based fabrication further offers the advantages of low cost, simplified processing, and facile compositional control. Previous reports have shown that solution-processed devices exhibit uniform performance, operational stability, and low energy consumption (Li et al., 2025). Building upon these advances, this work employs the mature and reliable oxide-TFT platform to develop the Otters spiking neuron.

### 2.2 TIME-TO-FIRST-SPIKE SNN

In contrast to rate-based encoding, which uses the frequency of spikes to represent information, TTFS encoding leverages the precise timing of a single spike. The core principle is that a stronger input stimulus causes a neuron's membrane potential to rise faster, reaching its firing threshold sooner. Thus, the information is encoded in the arrival time of the first—and only—spike within a given time window, $T$. This approach maximizes temporal sparsity and is highly efficient, as each neuron fires at most once (Che et al., 2024a).

The operation of a standard TTFS neuron involves two phases. First, the neuron integrates incoming spikes, updating its membrane potential $V_j^l(t)$. Second, it compares this potential to a firing threshold $\theta^l(t)$. A spike is generated at the first time step $t$ where the potential meets or exceeds the threshold:

$$s_j^l(t) = \begin{cases} 1, & \text{if } V_j^l(t) \geq \theta^l(t) \\ 0, & \text{otherwise} \end{cases} \tag{1}$$

However, the asynchronous nature of SNNs, combined with the "fire-as-early-as-possible" objective of TTFS, can lead to another problem. If a presynaptic neuron fires after a postsynaptic neuron has already fired, its spike becomes invalid for membrane potential accummalation. To solve this, we employ a Dynamic Firing Threshold (DFT) model that enforces a synchronous, layer-by-layer processing schedule (Wei et al., 2023). The threshold for any neuron in layer $l$ is set to infinity outside of a designated time window, effectively ensuring that layer $l$ is only active from time $T \cdot l$ to $T \cdot (l+1)$:

$$\theta^l(t) = \begin{cases} \theta_{\text{dynamic}}^l(t), & \text{if } T \cdot l \leq t \leq T \cdot (l+1) \\ +\infty, & \text{otherwise} \end{cases} \tag{2}$$

This scheduling guarantees that all spikes from a preceding layer are processed before the current layer can fire, thus preserving the valid causal relationship.

## 3 METHODS

This section details the methodology behind Otters. We first describe the core of our model: the optoelectronic synapse and the neuron model that performs the TTFS computation (Section 3.1). We then explain how these components are assembled into the complete optimized spiking transformer architecture (Section 3.2), and then, we outline the QNN-to-SNN conversion pipeline used to build the Otters model (Section 3.3). Finally, we present the framework for the comprehensive energy analysis used to validate our model's efficiency (Section 3.4).

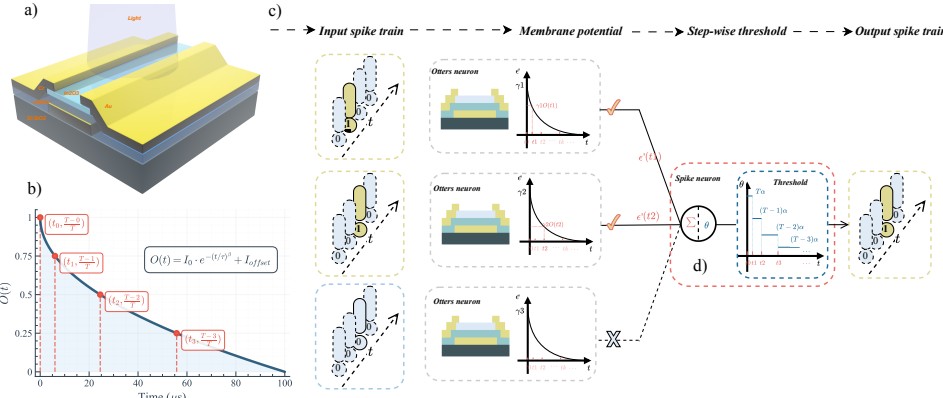

Figure 2: Device and workflow: (a) the custom-fabricated $In_2O_3$ thin-film transistor (TFT); (b) measured decay curve of the device response; (c) Otters neuron workflow.

### 3.1 OTTERS SPIKING NEURON

The core of our method is the Otters optoelectronic synapse, a hardware element that physically implements the time-modulated synaptic dynamics required for TTFS computation. Each synapse is composed of two main parts: a custom-fabricated Indium Oxide Thin-Film Transistor that provides a physical signal decay, and an analog-to-digital converter (ADC) that map and scale the analog signal to digital. Fabrication details for the device in Fig. 2(a) are provided in Appendix A.1. To ensure a deterministic response, the TFT is operated with a fixed light intensity, yielding a consistent non-linear decay curve, modeled by the function $O(t) = I_0 \cdot e^{-(t/\tau)^\beta} + I_{\text{offset}}$ (Li et al., 2024a; Liang et al., 2022) . We fit the model parameters by minimizing the sum of squared residuals using the differential evolution algorithm, yielding: $I_0 = 110.989$, $\tau = 1.3425$, $\beta = 0.495$, and $I_{\text{offset}} = -109.989$, showing in Figure 2(b). Thus, the TFT's current decay over time naturally forms the temporal component of the post-synaptic potential (PSP).

However, the device's physical nonlinearity poses a critical design challenge. For our QNN-to-SNN conversion to be lossless, the information encoded by a spike's timing must map to a set of uniformly spaced logical values. Specifically, a spike occurring at a physical time $t_k$ must represent the quantized value $(T - k)/T$. The device's non-linear decay, $O(t)$, means that the physical times $t_k$ at which the device output naturally equals these target values are themselves non-uniformly spaced. A naive approach using a constant threshold and uniform time sampling would therefore fail to establish the required functional equivalence.

Therefore, our solution is to reconcile the non-linear device physics with the linear encoding requirement. Instead of implementing a complex, non-uniform clock, we engineer a dynamic, step-wise decreasing firing threshold, $\theta^l(t)$, while operating the system on a standard, uniform physical clock. This threshold is designed to change its value only at the pre-calculated time points $\{t_k\}$ which are derived from the inverse of the physical decay function. This design ensures that the firing condition where the membrane potential exceeding the threshold can only be met at one of these discrete moments $t_k$. The neuron fires at the first such time point where its accumulated potential is sufficient. Consequently, the output spike time $t_k$ can encode the intended quantized value $(T - k)/T$. Because this encoding scheme is applied consistently throughout the network, the output spike of one layer provides a correctly timed and valued input to the next, ensuring the integrity of information

propagation across the entire architecture. The full mathematical formulation of this threshold will be defined in our conversion methodology in Section 3.3.

The ADC implements a $\gamma_{ij}^l$ times scaling mapping from the physical decay to a digital post-synaptic potential (PSP). Thus, the full PSP $\epsilon'$ generated by a presynaptic spike is therefore the direct analog output of the synapse at the time of arrival:

$$\epsilon'(t) = \gamma_{ij}^l \cdot O(t) \tag{3}$$

The membrane potential $V_j^l(t)$ of neuron j accumulates these PSPs. At each discrete physical time step t, the potential is updated based on incoming spikes:

$$V_j^l(t) = V_j^l(t-1) + \sum_{i \text{ s.t. } s_i^{l-1}(t)=1} \epsilon'(t) \tag{4}$$

A neuron fires when its membrane potential first meets or exceeds the dynamic threshold. This spike time, $t_{\text{spike},j}^l$, corresponds to the first logical timestep k where the condition is met:

$$t_{\text{spike},j}^l = \min\{t_k | V_j^l \geq \theta^l(t_k)\} \tag{5}$$

In adherence with the TTFS paradigm, the neuron is deactivated after firing to ensure at most one spike per inference cycle. Algorithm 1 in appendix formally describes this forward pass.

## 3.2 NETWORK STRUCTURE

A primary challenge in creating a spiking transformer is the matrix multiplication required for self-attention score calculation ($Q \cdot K^T$). While some rate-coded SNNs can simplify this by treating one matrix as a binary spike train (turning multiplication into selective addition), this approach is incompatible with TTFS encoding, which requires decoding spikes into non-binary values. To overcome the problem, we quantize the key ($K$) and value ($V$) projections to a single bit, $\{+1, -1\}$. Consequently, the dot product with a TTFS-encoded query ($Q$) is computed using only selective, additions and subtractions. This allows us to eliminate the multiplication bottleneck while still benefiting from the high sparsity of TTFS. To implement this 1-bit attention mechanism efficiently, we designed a supporting dataflow architecture inspired by the Canon architecture (Bai et al., 2025), as illustrated in Figure 3. During inference, the binary K (or V) vectors are pre-loaded into the local memory of a Processing Element (PE) array. The TTFS-encoded input stream (representing Q) is broadcast to the PEs. As shown, each PE computes a partial sum by accumulating its local K values only at the time steps corresponding to incoming spikes. These partial sums are then passed between PEs for final accumulation. This architecture minimizes data movement and leverages the spatio-temporal sparsity of the TTFS input for energy efficiency.

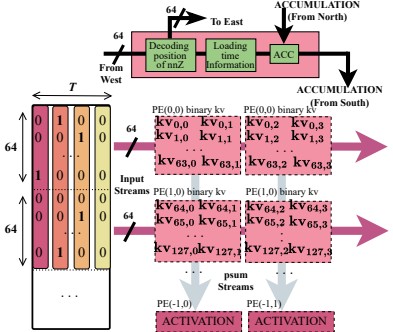
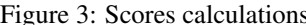
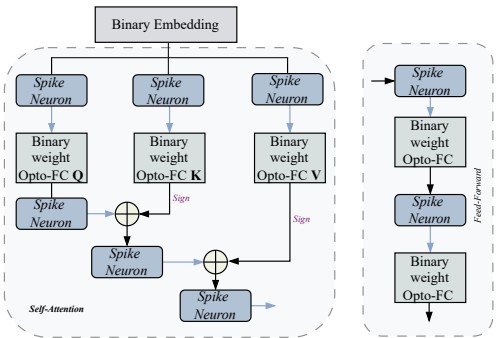

Figure 3: Scores calculations          Figure 4: Otters-based transformer structure

This self-attention module, along with feed-forward layers built from our Opto-FC and spiking neuron primitives, forms the complete Otters transformer architecture shown in Figure 4.

### 3.3 QNN-TO-SNN CONVERSION

To overcome the challenges of direct SNN training, we employ a QNN-to-SNN conversion methodology. We first train a QNN and then map its learned parameters to an equivalent Otters SNN, ensuring the two networks are functionally identical. To formalize this relationship, we first define the specific QNN layer architecture that enables this equivalence. A compatible QNN layer computes its output $x_{q,j}^l$ for neuron j as follows:

$$a_j^l = \sum_i w_{ij}^l x_{q,i}^{l-1} + b_j^l \tag{6}$$

$$x_{q,j}^l = Q(a_j^l) = \alpha^l \cdot \text{Clip}(\lfloor \frac{a_j^l}{\alpha^l} \rfloor, 0, 2^n - 1) \tag{7}$$

where $a_j^l$ is the pre-activation, $w_{ij}^l$ are the weights, $b_j^l$ is the bias, and $\alpha^l$ is the quantization scaling factor for layer l. With this structure established, we can now state the proposition that governs the exact conversion from the trained QNN to the Otters SNN:

**Proposition 1.** *An Otters SNN layer (as defined in Section 3.1) is functionally equivalent to a trained n-bit QNN layer (as defined above) if its parameters are constructed as follows:*

1. *The number of discrete time steps in the SNN simulation window, T, is set to match the number of positive quantization levels of the n-bit QNN: $T = 2^n - 1$.*

2. *The mapping from a logical time step $k \in \{0, 1, \ldots, T-1\}$ to a physical spike time $t_k$ is defined such that the device's output at that instant, $O(t_k)$, is linearly proportional to the remaining time in the window: $O(t_k) = \frac{T-k}{T}$.*

3. *The SNN's physical scaling factor for the synapse connecting neuron i in layer $l-1$ to neuron j in layer l, $\gamma_{ij}^l$, is set based on the corresponding QNN weight and the quantization scale of the previous layer: $\gamma_{ij}^l = w_{ij}^l \cdot \alpha^{l-1} \cdot T$.*

4. *The SNN's firing threshold for neuron j in layer l is a step-wise decreasing function of time, defined as: $\theta^l(t) = \alpha^l \cdot (T - k)$, for $t_k \leq t < t_{k+1}$.*

The proof proceeds in two steps. First, during integration, we show that the accumulated membrane potential $V_j^l$ in an SNN neuron is numerically identical to the corresponding QNN neuron's pre-activation value $a_j^l$. We binarize weights $w_{ij}^l$ to reduce energy and increase the reuse of factor $\gamma_{ij}^l$. Second, during firing, we show that the engineered, time-dependent threshold $\theta_j^l(t)$ compensates for the device's intrinsic non-linearity by permitting firing only at pre-calculated time points, $t_k$, derived from the inverse of the physical decay function. This mechanism ensures the SNN neuron fires at a time $t_{\text{spike},j}^l$ that precisely encodes the QNN's quantized output $x_{q,j}^l$. A detailed proof is provided in the Appendix A.2.

### 3.4 ENERGY ANALYSIS

To evaluate the efficiency of our approach, we formulate an analytical energy model. The total inference energy ($E$) is decomposed into three primary components: computation energy ($E_{\text{Compute}}$), data movement energy ($E_{\text{Data}}$), and analog energy ($E_{\text{Analog}}$) (Yan et al., 2024; Dampfhoffer et al., 2022). The computation energy, $E_{\text{Compute}}$, accounts for arithmetic operations of additions. The data movement energy, $E_{\text{Data}}$, encompasses the energy for transferring data, including both dynamic and static power consumption. The final component, $E_{\text{Analog}}$, includes the energy to power the TFT, the sampling energy, and ADC energy required to converted the analog signal to scaled digital value, collectively represented as $E_{\text{Analog}}^{\text{Read}}$. Thus, we have $E = E_{\text{Compute}} + E_{\text{Data}} + E_{\text{analog}}$. For our energy calculation, we consider a spatial dataflow architecture where information (e.g., spike packets) is communicated over a Network-on-Chip (NoC) (Yan et al., 2024). This architecture is representative of modern specialized hardware such as neuromorphic chips like Loihi (Lines et al., 2018) and dataflow AI accelerators like Tenstorrent (Vasiljevic et al., 2021) and Sambanova (Prabhakar et al., 2022). We consider the control logic energy to be negligible as our analysis focuses on specialized accelerator designs where such overhead is minimal (Yan et al., 2024).

Thus, we model the energy consumption of our Otters-based linear projections and attention operations. We disregard the computational cost of certain operations, such as Softmax and Layer Normalization, as their contribution to the total compute is negligible compared to large-scale matrix multiplications. The energy to perform a linear projection, $E_{\text{Opto-FC}}$, is modeled by Equation 8. The calculation is performed over a batch of size $B$, a sequence of length $S$, and for $C_o$ output channels. The total energy is the sum of the following components per output element:

$$E_{\text{Opto-FC}} = \underbrace{B \cdot S \cdot C_o}_{\text{Total Outputs}} \cdot \Bigg( \underbrace{C_i \cdot T \cdot \big(s_r \cdot (E_{\text{ACC}} + E_{\text{Analog}}^{\text{Read}} + E_{\text{move}}^{\text{sparse}})}_{\text{Spike Processing}} + E_{\text{leakage}}\big) + \underbrace{T \cdot (E_{\text{CMP}} + E_{\text{threshold}}^{\text{Read}})}_{\text{Thresholding}} + \underbrace{E_{\text{binarykv}}^{\text{Write}}}_{\text{K/V Write}} \Bigg) \quad (8)$$

- Computation Energy: This includes the energy for sparse accumulations. The total number of active accumulate operations is the workload ($C_i$) scaled by the average number of spikes ($T \cdot s_r$), with each operation costing $E_{\text{ACC}}$. Additionally, each of the $C_o$ output neurons performs $T$ comparisons against its threshold, costing $E_{\text{CMP}}$ per comparison.

- Data Movement Energy: This is composed of dynamic and static costs. Dynamic energy ($E_{\text{move}}^{\text{sparse}}$) is consumed to move spike data and is proportional to the spike rate ($s_r$). Otters paradigm requires a dynamic, step-wise decreasing threshold, which introduces additional dynamic energy of reading operations $E_{\text{threshold}}^{\text{Read}}$ which is need for spiking neurons. Static energy ($E_{\text{leakage}}$) accounts for constant leakage power over the time window $T$. After the computation completes, an additional energy cost, $E_{\text{binarykv}}^{\text{Write}}$, is incurred to write the generated binary Key and Value vectors to SRAM, as described in Section 3.2.

- Analog Energy: Each incoming spike initiates two analog operations: first, the optoelectronic synapse emits light, and second, the decay function is sampled at a specific time $t_k$. The energy for both the light emission and the sampling event is calculated by integrating the instantaneous power ($P = V \cdot I$) over the duration of each respective operation. For the readout circuitry, including the amplifier and the look-up-table, we adopt the energy values of a successive approximation register-assisted pipelined ADC (Su et al., 2023). The total energy consumed in this process is denoted $E_{\text{Analog}}^{\text{Read}}$ per spike.

The energy model for attention score calculation, $E_{\text{Opto-score}}$, is analogous to the linear projection, with two key differences which lies in the outer dimensions and an additional data movement cost for reading the binary Key (or Value) vector from SRAM to determine whether the sampled membrane potential should be added to or subtracted from the accumulator, showing in Appendix A.3.

## 4 RESULTS

In this section, we evaluate Otters on seven datasets from the GLUE benchmark. We compare its performance against both standard QNN and SNN baselines, using BERT$_{\text{base}}$ as the teacher model for knowledge distillation. We further provide a detailed analysis of the model's energy efficiency and robustness. All experiments were conducted on three NVIDIA A100 GPUs with a fixed 4-bit simulation window size recommended by Sorbet (Tang et al., 2025) (In our setting, it is equal to timestep $T = 15$). The training process of Otters is shown in Appendix A.5 and the description of dataset we adapted is shown in Appendix A.6.

### 4.1 GLUE BENCHMARK PERFORMANCE

As shown in Table 1, Otters achieves SOTA results among SNNs across all seven evaluated GLUE tasks, consistently outperforms larger and more complex SNN models like SpikingBERT and SpikeLM. For example, Otters surpasses existing SNNs and achieve an accuracy of 68.95% on RTE and 91.28% on SST-2. The average accuracy for Otters is 83.22%, which is 3.42% and 2.98% higher than Sorbet and SpikeLM, respectively.

Table 1: Performance comparison on the GLUE benchmark. All scores are accuracy, except for STS-B (Pearson correlation). "*" indicates that the model size was not reported in the original paper. **Bold** indicates the best performance among SNN models. Only Otters quantizes KV to 1 bit.

| Model | Size | QQP | MNLI-m | SST-2 | QNLI | RTE | MRPC | STS-B | Average |
|---|---|---|---|---|---|---|---|---|---|
| BERT$_{base}$ (Devlin et al., 2019) | 418M | 91.3 | 84.7 | 93.3 | 91.7 | 72.6 | 88.2 | 89.4 | 87.31 |
| DistilBERT (Sanh et al., 2020) | 207M | 88.5 | 82.2 | 91.3 | 89.2 | 59.9 | 87.5 | 86.9 | 83.64 |
| TinyBERT$_6$ (Jiao et al., 2020) | 207M | - | 84.6 | 93.1 | 90.4 | 70.0 | 87.3 | 83.7 | 84.85 |
| Q2BERT (Zhang et al., 2020) | 43.0M | 67.0 | 47.2 | 80.6 | 61.3 | 52.7 | 68.4 | 4.4 | 54.51 |
| BiT (Liu et al., 2022) | 13.4M | 82.9 | 77.1 | 87.7 | 85.7 | 58.8 | 79.7 | 71.1 | 77.57 |
| SpikingFormer (Zhou et al., 2023) | * | 83.8 | 67.8 | 82.7 | 74.6 | 58.8 | 74.0 | 72.3 | 73.43 |
| SpikingBERT (Bal & Sengupta, 2024) | 50M | 86.8 | 78.1 | 88.2 | 85.2 | 66.1 | 79.2 | 82.2 | 80.83 |
| SpikeLM (Xing et al., 2024b) | * | 87.9 | 76.0 | 86.5 | 84.9 | 65.3 | 78.7 | 84.3 | 80.51 |
| 1-bit SpikeLM (Xing et al., 2024b) | * | 87.2 | 74.9 | 86.6 | 84.5 | 65.7 | 78.9 | 83.9 | 80.24 |
| 1-bit Sorbet (Tang et al., 2025) | 13.4M | 86.5 | 77.3 | 90.4 | 86.1 | 60.3 | 79.9 | 78.1 | 79.80 |
| **Otters (Ours)** | 13.4M | **87.67** | **78.50** | **91.28** | **86.42** | **68.95** | **84.56** | **85.19** | **83.22** |

## 4.2 ENERGY EFFICIENCY

We analyzed the energy consumption of Otters on the SST-2 dataset, comparing it against full-precision and quantized BERT models we converted from, as well as SOTA 1-bit SNNs. The 1-bit quantized BERT is the QNN Otters converted from which sharing the same structure and parameters. As detailed in Table 2, Otters consumes only 4.06 mJ per inference for one attention block. This represents a 41.36× energy saving compared to the full BERT$_{base}$ model and a 2.72× efficient compared to the 1-bit quantized BERT. Otters is also more efficient than previous SNNs, taking 3.04× energy saving of Sorbet and 1.77× of SpikingLM.

Appendix A.3 provides the full energy equation, detailed measurements for all compared models, and an ablation study comparing traditional TTFS with Otters. All energy figures include compute, data movement, and static components. In the same appendix section, we provide a detailed breakdown of the energy consumption in Otters to clarify the contribution of each component.

Table 2: Energy consumption analysis on the SST-2 dataset. Energy is reported for each FC layer (QKV linear projections), each QKV self-attention score calculation, and the total sum per inference. The Energy Ratio is Energy(Full BERT) / Energy(Model) (↑ higher is better).

| Model | FC (mJ) | QKV (mJ) | Total (mJ) | Energy Ratio(↑) |
|---|---|---|---|---|
| Full BERT (Devlin et al., 2019) | 50.35 | 8.41 | 167.92 | 1.00x |
| 1-bit Quantized BERT | 3.31 | 0.55 | 11.03 | 15.2x |
| Sorbet (Tang et al., 2025) | 3.39 | 1.08 | 12.34 | 13.61x |
| SpikingBERT (Bal & Sengupta, 2024) | 6.37 | 2.05 | 23.22 | 7.23x |
| SpikingLM (Xing et al., 2024b) | 2.09 | 0.46 | 7.2 | 23.32x |
| Otters (4bit kv) | **1.14** | **0.53** | **4.49** | **37.40x** |
| **Otters (1bit kv)** | **1.14** | **0.33** | **4.06** | **41.36x** |

## 4.3 EFFECT OF KV CACHE QUANTIZATION

To further optimize energy, we explored the impact of quantizing the Key and Value projections in the self-attention mechanism. Table 3 shows that reducing the KV precision from 4-bit to 1-bit (Otters-1bitkv) yields a 10% reduction in total energy consumption (from 4.49 mJ to 4.06 mJ). This energy saving comes at the cost of 0.23% drop in accuracy on SST-2, demonstrating a highly favorable trade-off between efficiency and performance.

Table 3: Impact of KV quantization on energy and accuracy on SST-2.

| Model | Energy (mJ) | | | Accuracy (%) |
|---|---|---|---|---|
| | FC | QKV | Total | |
| Otters-4bitkv | 1.14 | 0.53 | 4.49 | 91.51 |
| Otters-1bitkv | 1.14 | 0.33 | 4.06 | 91.28 |

## 4.4 ROBUSTNESS DISCUSSION TO HARDWARE PARAMETER VARIATIONS

The practical deployment of the Otters paradigm hinges on the assumption that the physical decay characteristics of all optoelectronic synapses are uniform. However, analog hardware is inevitably subject to device-to-device variability from the fabrication process, which is a significant challenge for large-scale integration (Garg et al., 2022). Such inherent hardware noise can corrupt the precise time-to-value mapping that underpins our conversion method. To evaluate robustness against hardware variability (Fagbohungbe & Qian, 2021; Xuan & Narayanan, 2022; Su et al., 2024), we injected zero-mean standard deviation Gaussian noise, which is commonly used to simulate and represent hardware noise, into the physical decay function, $O(t)$, and its parameters, $\tau$ and $\beta$. The injected noise is proportional to parameter magnitude: at level $k$, each parameter $p$ is scaled as $p \leftarrow p\,(1 \pm k)$. As shown in Table 4, the baseline Otters model can tolerate about $5\%$ total output $O(t)$ difference with minimal accuracy impact and remains robust.

To keep improving the robustness, we propose Hardware-Aware Training (HAT), a method that builds robustness by simulating hardware non-idealities during training. We introduce two variants, $\text{HAT}^1$ and $\text{HAT}^2$, by injecting 10% and 20% Gaussian noise, respectively, into the QNN's activations (see Eq. 7). As shown in Table 4, both HAT settings improve noise resilience. The $\text{HAT}^2$ model maintains a stable accuracy of 80.8% even under a 20% noise level. $\text{HAT}^2$, trained with more noise, excels in high-noise regimes, whereas $\text{HAT}^1$ achieves higher accuracy in low-noise conditions while still substantially outperforming the baseline (e.g., a 11.5% accuracy gain with 12% noise in $O(t)$). Thus, HAT-trained models trade a minor drop in peak accuracy for a significant increase in resilience against hardware noise. This demonstrates that the model can be regularized to generalize across a range of hardware imperfections. Consequently, the HAT noise level can be tuned to the manufacturing tolerances of a specific hardware platform, ensuring reliable real-world performance and validating the Otters paradigm as a robust and practical approach.

Table 4: Noise Robustness in Physical Decay Function. The best performance per noise level within each component group is in boldface. Results: mean $\pm$ std dev over 3 runs.

| Method | Gaussian Noise Level | | | | |
|---|---|---|---|---|---|
| **Full Function $O(t)$ Experiments** | | | | | |
| $O(t)$ | 0.04 | 0.08 | 0.12 | 0.16 | 0.20 |
| Otters | **89.9 ± 0.8** | 86.1 ± 0.8 | 73.8 ± 0.9 | 58.0 ± 0.8 | 53.3 ± 0.4 |
| Otters+$\text{HAT}^1$ | 89.3 ± 0.5 | **89.0 ± 0.6** | 85.3 ± 0.6 | 76.5 ± 0.4 | 61.0 ± 0.8 |
| Otters+$\text{HAT}^2$ | 87.4 ± 0.5 | 87.2 ± 0.7 | **85.9 ± 0.7** | **85.2 ± 0.7** | **80.8 ± 0.3** |
| **Parameter $\beta$ Experiments** | | | | | |
| $\beta$ | 0.01 | 0.02 | 0.03 | 0.04 | 0.05 |
| Otters | **90.2 ± 0.1** | **89.5 ± 0.1** | 87.5 ± 1.8 | 79.2 ± 3.4 | 72.5 ± 1.4 |
| Otters+$\text{HAT}^1$ | 89.6 ± 0.1 | 89.2 ± 0.3 | **89.1 ± 0.4** | **87.7 ± 1.2** | 80.6 ± 2.8 |
| Otters+$\text{HAT}^2$ | 87.8 ± 0.7 | 88.1 ± 0.4 | 87.4 ± 1.0 | 85.5 ± 1.1 | **83.8 ± 0.5** |
| **Parameters $\tau$ Experiments** | | | | | |
| $\tau$ | 0.10 | 0.20 | 0.30 | 0.40 | 0.50 |
| Otters | **90.3 ± 0.1** | **90.1 ± 0.5** | 87.0 ± 1.7 | 75.0 ± 4.1 | 67.5 ± 4.8 |
| Otters+$\text{HAT}^1$ | 89.6 ± 0.5 | 89.3 ± 1.2 | **88.2 ± 0.7** | **88.2 ± 1.0** | 76.1 ± 2.8 |
| Otters+$\text{HAT}^2$ | 87.8 ± 0.6 | 87.7 ± 0.8 | 87.1 ± 0.6 | 87.2 ± 0.3 | **83.0 ± 1.4** |

## 5 DISCUSSION AND FUTURE WORKS

To make SNNs more energy-efficient, Otters focuses on hardware-software co-design to optimize a core computing operation. We replace the temporal decay function and its following multiplication by sampling the natural decay of an oxide optoelectronic synapse. This shifts the computation from digital to analog for a more energy-efficient computing method. However, at the same time, many related works are also working toward better energy efficiency for SNNs, but from another direction, such as algorithmic and architectural improvements. These methods, including QKFormer, SSSA, and $\text{A}^2\text{OS}^2\text{A}$, optimize the spiking attention mechanism from $O(N^2)$ to linear complexity (linear-attention SNNs) (Zhou et al., 2024; Wang et al., 2025b; Guo et al., 2025).These two directions are

not in conflict. They are complementary and solve different parts of the problem. Future work includes designing a linear attention mechanism based on the Otters spiking neuron to make energy use even more efficient.

We further discuss the hardware scale. The $In_2O_3$ optoelectronic synapse in our prototype has an effective area of about 0.012 mm$^2$ with a channel length of 30 $\mu$m. This size is mainly due to the precision limits of our fabrication equipment and does not represent the scaling limits of $In_2O_3$ technology. Since the goal of this paper is to validate the optoelectronic TTFS mechanism, we did not focus on device size optimization. However, recent work has demonstrated $In_2O_3$ transistors with channel lengths down to 40 nm (Si et al., 2021), showing that the device area can be reduced by nearly three orders of magnitude. Additionally, several studies show that oxide devices can be integrated in 3D stacked layers, providing another path to further reduce the effective area per synapse (Tang et al., 2022; Yuvaraja et al., 2024; Kwak et al., 2024). With the TTFS mechanism verified, we plan to focus on device scaling in future work. We also note that the device count can be reduced by device sharing. For example, consider a Q-projection layer with input channels $C_i$ and output channels $C_o$. If every weight required one device, the layer would need $O(C_i \times C_o)$ devices. In the Otters setting, this number would reach $10^6$–$10^7$, which is clearly not practical. However, in our setting (introduced in Section 3.3), all weights are quantized to 1-bit. This means the decay function is identical for the entire layer, making it possible to share the $In_2O_3$ device. In the best case, one FC layer can reuse just one physical device. Thus, a single BERT block would require about 8 devices, resulting in a total area of 0.096 mm$^2$: 6 for the self-attention module (projections of Q, K, V, their multiplication and output) and 2 for the feed-forward module. (In real systems, the number of devices might be higher, and sharing may be limited by architecture and compiler constraints.) The next step of this work is to design an architecture that increases this device reuse to further reduce the chip area.

## 6  CONCLUSION

This paper introduces Otters, a new paradigm for energy-efficient neuromorphic computing that challenges this digital-centric approach. Through a hardware-software co-design, we repurpose the natural signal decay of a custom-fabricated optoelectronic synapse, transforming this physical phenomenon into a computational method. This allows us to eliminate the costly evaluation of the decay function inherent in traditional TTFS, thereby fusing computation and memory into the physical process. To deploy this paradigm in complex architectures such as transformers, we developed a QNN-to-SNN conversion algorithm that circumvents the challenges of direct SNN training. The Otters model achieves state-of-the-art accuracy among SNNs on seven GLUE benchmark datasets, while simultaneously delivering a 1.77$\times$ improvement in energy efficiency over previous leading spiking models. By directly harnessing fundamental device physics for computation, this work demonstrates a new path to a more energy-efficient neuromorphic computing design.

## 7  ACKNOWLEDGEMENT

This research is partially supported by the Ministry of Education, Singapore, under the Academic Research Fund Tier 1 (FY2024).

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

## A APPENDIX

### A.1 FABRICATED IN$_2$O$_3$ TFTS

To prepare the indium oxide thin-film transistors (In$_2$O$_3$ TFTs), indium nitrate was first dissolved in a mixed solvent of 2-methoxyethanol (2-ME), acetylacetone (AcAc), and ammonium hydroxide (NH$_3$·H$_2$O) to form a precursor solution, which was stirred overnight to ensure complete dissolution and coordination. Subsequently, gate electrodes were fabricated on a silicon substrate coated with a SiO$_2$ insulating layer, followed by sequential deposition of 8 nm chromium and 50 nm gold via electron-beam evaporation. A 30 nm-thick Al$_2$O$_3$ dielectric layer was then uniformly deposited over the substrate using atomic layer deposition (ALD). The In$_2$O$_3$ precursor solution was spin-coated onto the dielectric surface, after which the channel regions were defined through standard photolithography, and the unprotected areas were removed by hydrochloric acid wet etching. The films were annealed in air at 300 °C for 1 hour to enhance crystallinity and improve film quality. Portions of the Al$_2$O$_3$ layer were subsequently etched to expose selected regions of the gate electrodes. Finally, source and drain electrodes, along with interconnects, were patterned and metallized with an additional 8 nm chromium and 50 nm gold layer via electron-beam evaporation. The indium oxide thin-film transistor was characterized under a gate bias of 0 V and a drain bias of 5 mV. Upon 405 nm laser illumination, oxygen vacancies in the channel layer were photoionized, generating free electrons and thereby enhancing the channel conductivity.

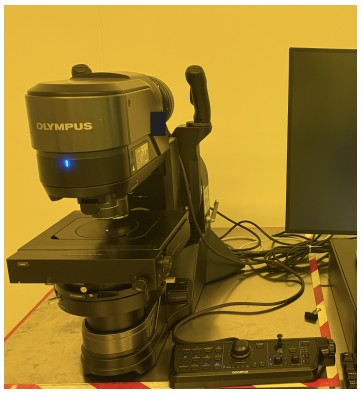
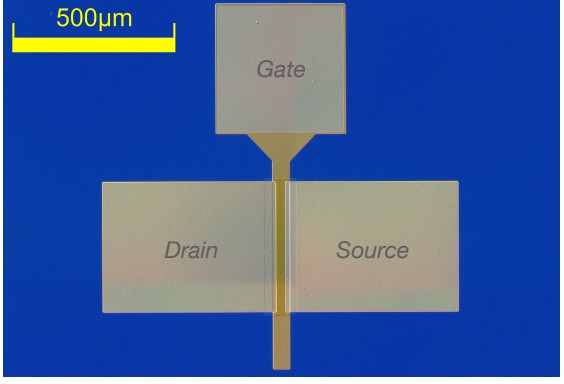

(a) Optical microscope        (b) Otters under optical microscope

Figure 5: Details design of Otters

Regarding integration with state-of-the-art digital hardware, recent studies show that In$_2$O$_3$ TFTs are compatible with 3D stacking on CMOS chips. This is because they have a low thermal budget ($\leq$ 300 °C) and good uniformity (Tang et al., 2022). This provides a practical path for vertical integration. However, several challenges remain, such as managing wire density in stacked layers, aligning the photodetectors/TFT layers with metal layers, and handling device variability in large arrays.

### A.2 PROOF FOR PROPOSITION 1

#### A.2.1 INTEGRATION PHASE EQUIVALENCE

The foundation of the proof lies in the relationship between the QNN's discrete value and the SNN's spike time. From Section 3.3, we know that the QNN output activation from the previous layer is $x_{q,i}^{l-1} = \alpha^{l-1} \cdot q_i^{l-1}$, where $q_i^{l-1}$ is the integer value:

$$q_i^{l-1} = \text{Clip}\left( \left\lfloor \frac{a_i^{l-1}}{\alpha^{l-1}} \right\rfloor, 0, T \right) \tag{9}$$

We define the TTFS encoding scheme such that this integer value $q_i^{l-1}$ is represented by a single spike from neuron $i$ at the discrete time step $k$:

$$k = T - q_i^{l-1} \tag{10}$$

This encoding adheres to the TTFS principle: a larger integer value $q_i^{l-1}$ results in a smaller time step $k$, signifying an earlier spike. A value of 0 corresponds to no spike within the active window, and the maximum value $T$ corresponds to a spike at $k = 0$.

When a presynaptic neuron $i$ fires at time step $k$, its contribution to the postsynaptic potential of neuron $j$ is given by the *Otters* PSP function, $\epsilon'(\cdot)$. Using the conditions specified in Proposition 1, the normalized value produced by the decay function $O(t)$ at time step $t_k$ is:

$$O(t_k) = \frac{T - k}{T} \tag{11}$$

Substituting the encoding relationship from Step 1 ($k = T - q_i^{l-1}$):

$$O(t_k) = \frac{T - (T - q_i^{l-1})}{T} = \frac{q_i^{l-1}}{T} \tag{12}$$

Thus we find that the normalized output of the physical decay process at the spike time $t_k$ is directly proportional to the integer value $q_i^{l-1}$ it is meant to encode.

The full PSP contribution from the synapse connecting $i$ to $j$ is the product of this normalized value and the scaling factor $\gamma_{ij}^l$. Using the definition of $\gamma$ from Proposition 1:

$$\gamma_{ij}^l = w_{ij}^l \cdot \alpha^{l-1} \cdot T \tag{13}$$

The PSP is therefore:

$$\begin{aligned}
\epsilon'(w_{ij}^l, t_k) &= \gamma_{ij}^l \cdot O(t_k) \\
&= (w_{ij}^l \cdot \alpha^{l-1} \cdot T) \cdot \left(\frac{q_i^{l-1}}{T}\right) \\
&= w_{ij}^l \cdot (\alpha^{l-1} \cdot q_i^{l-1})
\end{aligned} \tag{14}$$

Recognizing that $x_{q,i}^{l-1} = \alpha^{l-1} \cdot q_i^{l-1}$, we find:

$$\epsilon'(w_{ij}^l, t_k) = w_{ij}^l \cdot x_{q,i}^{l-1} \tag{15}$$

This shows that the contribution of a single spike in the SNN is exactly equal to the weighted input term in the QNN.

The final membrane potential $V_j^l$ is the sum of all such PSPs from incoming spikes, plus the bias term:

$$V_j^l = \sum_i \epsilon'(w_{ij}^l, t_{\text{spike},i}^{l-1}) + b_j^l = \sum_i (w_{ij}^l \cdot x_{q,i}^{l-1}) + b_j^l \tag{16}$$

By comparing this with the definition of the QNN pre-activation from Section 1.2,

$$a_j^l = \sum_i w_{ij}^l x_{q,i}^{l-1} + b_j^l \tag{17}$$

we arrive at the desired equality:

$$V_j^l = a_j^l \tag{18}$$

### A.2.2   FIRING PHASE EQUIVALENCE

In this section, we prove that the integer value encoded by the SNN's output spike time, $t_{\text{spike},j}^l$, is equal to the integer value of the QNN's output, $q_j^l$. That is, if $t_{\text{spike},j}^l$ corresponds to time step $k_{\text{fire}}$, we have:

$$T - k_{\text{fire}} = q_j^l = \text{Clip}\left(\left\lfloor \frac{a_j^l}{\alpha^l} \right\rfloor, 0, T\right) \tag{19}$$

According to the SNN model definition, the neuron fires at the earliest discrete time step $k$ for which its potential $V_j^l$ meets or exceeds the threshold $\theta^l(t_k)$.

$$V_j^l \geq \theta^l(t_k) \tag{20}$$

Substituting the result from Part I ($V_j^l = a_j^l$) and the definition of the time-varying threshold from Proposition 1 ($\theta^l(t_k) = \alpha^l \cdot (T - k)$), the firing condition becomes:

$$a_j^l \geq \alpha^l \cdot (T - k) \tag{21}$$

Assuming $\alpha^l > 0$, we can rearrange the inequality to solve for the term $(T - k)$, which represents the integer value that would be encoded by a spike at time step $k$:

$$\frac{a_j^l}{\alpha^l} \geq T - k \tag{22}$$

The threshold $\theta^l(t_k) = \alpha^l(T - k)$ is a monotonically decreasing function of the time step $k$. For a fixed membrane potential $a_j^l$, this means that if the firing condition is met for a certain time step $k^*$, it will also be met for all subsequent time steps $k > k^*$. The TTFS firing rule dictates that the neuron fires at the first time step that satisfies the condition. This corresponds to finding the smallest integer $k$ that satisfies the inequality (largest $T - k$ ). By the definition of the floor function, the integer value encoded by the output spike, which is defined by our encoding scheme as $q_{\text{out},j}^l = T - k_{\text{fire}}$, is:

$$q_j^l = \left\lfloor \frac{a_j^l}{\alpha^l} \right\rfloor \tag{23}$$

The derivation above assumes the result of the floor function falls within the valid range of encodable integers. We now analyze the boundary conditions imposed by the finite simulation window.

- Upper Bound (Clipping at T) If the pre-activation $a_j^l$ is very large such that $\lfloor a_j^l/\alpha^l \rfloor > T$, the condition $a_j^l/\alpha^l \geq T - k$ will be satisfied for all $k \in T$. The neuron will fire at the earliest possible time step, which is $k = 0$. The value encoded by a spike at $k = 0$ is $T - 0 = T$. This naturally implements the upper bound of the clipping function, mapping any integer value greater than $T$ to $T$.

- Lower Bound (Clipping at 0). If the pre-activation $a_j^l$ is such that $\lfloor a_j^l/\alpha^l \rfloor < 0$ , then the term $a_j^l/\alpha^l$ is negative. Thus, the firing condition $a_j^l/\alpha^l \geq T - k$ can never be satisfied. The neuron will not fire within the time window. The absence of a spike is interpreted as encoding the integer value 0. This naturally implements the lower bound of the clipping function.

Combining these cases, the integer value encoded by the SNN's firing mechanism, $q_{\text{out},j}'$, is:

$$q_{\text{out},j}' = \text{Clip}\left( \left\lfloor \frac{a_j^l}{\alpha^l} \right\rfloor, 0, T \right) \tag{24}$$

This is identical to the definition of the QNN's integer output, $q_j^l$.

## A.3 ENERGY ANALYSIS

### A.3.1 ENERGY COMPARISON WITH RELATED WORKS

For the energy calculation in Table 2, we did not use the numbers reported in the original papers. Since energy depends heavily on batch size and sequence length, a direct comparison is often unfair. Furthermore, most related work does not account for hardware overhead, which dominates the total energy. Therefore, we re-calculated their energy to ensure a fair comparison. We assumed the same model size, batch size, sequence length and weight quantization level for all works.

For transformer baselines, the energy consumption of a full precision BERT (FP32) is dominated by expensive 32-bit multiply-accumulate (MAC) operations and data movement. We also compared

it to a quantized BERT (INT4), which uses the same architectural settings as Otter, including 1-bit weight, 4-bit activations and a 1-bit Key-Value (KV) cache.

For SNNs, we used the energy equations we built for them, which depend on the spike rate ($s_r$) and the number of timesteps ($T$). To match Otters' performance, we selected the most optimized settings for the baselines (Sorbet, SpikingBERT, and SpikingLM). This resulted in spike rates of 13% for Sorbet, 25% for SpikingBERT, and 33% for SpikingLM. We used the timestep values ($T$) directly from their original papers: 16, 16, and 4, respectively.

**Full BERT (FP32)**

$$E_{FC} = B \cdot S \cdot C_o \cdot (\gamma \cdot C_i \cdot (E_{\text{MAC}} + E_{\text{weight}}^{\text{Read}} + 32 \cdot E_{\text{move}}^{\text{sparse}}) + C_i \cdot E_{\text{leakage}} + 2E_{\text{clamp}} + E_{kv}^{\text{write}}) \quad (25)$$

$$E_{FC-score} = B \cdot h \cdot S^2 \cdot (d_k \cdot \gamma \cdot (E_{\text{kv}}^{\text{Read}} + E_{\text{MAC}} + 32 \cdot E_{\text{move}}^{\text{sparse}}) + d_k \cdot E_{\text{leakage}} + 2E_{\text{clamp}}) \quad (26)$$

**Quantized BERT**

$$E_{FC_q} = B \cdot S \cdot C_o \cdot (\gamma \cdot C_i \cdot (E_{\text{MAC}} + E_{\text{weight}}^{\text{Read}} + \log_2(T+1)E_{\text{move}}^{\text{sparse}}) + C_i \cdot E_{\text{leakage}} + 2E_{\text{clamp}} + E_{kv}^{\text{write}}) \quad (27)$$

$$E_{FC_q-score} = B \cdot h \cdot S^2 \cdot (d_k \cdot \gamma \cdot (E_{\text{kv}}^{\text{Read}} + E_{\text{MAC}} + \log_2(T+1) \cdot E_{\text{move}}^{\text{sparse}}) + d_k \cdot E_{\text{leakage}} + 2E_{\text{clamp}}) \quad (28)$$

**Typical SNNs**

$$E_{\text{SNN-FC}} = B \cdot S \cdot C_o \cdot (C_i \cdot s_r \cdot T \cdot (E_{\text{ACC}} + E_{\text{weight}}^{\text{Read}} + E_{\text{move}}^{\text{sparse}}) + C_i \cdot T \cdot E_{\text{leakage}} + T \cdot (E_{\text{CMP}} + s \cdot E_{\text{SUB}}) + E_{\text{kv}}^{\text{Write}}) \quad (29)$$

$$E_{\text{SNN-score}} = B \cdot h \cdot S^2 \cdot (d_k \cdot s_r \cdot T \cdot (E_{\text{kv}}^{\text{Read}} + E_{\text{ACC}} + E_{\text{move}}^{\text{sparse}}) + d_k \cdot T \cdot E_{\text{leakage}} + T \cdot (E_{\text{CMP}} + s \cdot E_{\text{SUB}})) \quad (30)$$

**Otters**

$$E_{\text{Opto-FC}} = \underbrace{B \cdot S \cdot C_o}_{\text{Total Outputs}} \cdot \bigg( \underbrace{C_i \cdot T \cdot \big( s_r \cdot (E_{\text{ACC}} + E_{\text{Analog}}^{\text{Read}} + E_{\text{move}}^{\text{sparse}})}_{\text{Spike Processing}}$$
$$+ E_{\text{leakage}} \big) + \underbrace{T \cdot (E_{\text{CMP}} + E_{\text{threshold}}^{\text{Read}})}_{\text{Thresholding}} + \underbrace{E_{\text{binarykv}}^{\text{Write}}}_{\text{K/V Write}} \bigg) \quad (31)$$

$$E_{\text{Opto-score}} = \underbrace{B \cdot h \cdot S^2}_{\text{Total Scores}} \cdot \bigg( \underbrace{d_k \cdot T \cdot \big( s_r \cdot (E_{\text{ACC}} + E_{\text{Analog}}^{\text{Read}} + E_{\text{move}}^{\text{sparse}} + E_{\text{binarykv}}^{\text{Read}})}_{\text{Spike Processing}}$$
$$+ E_{\text{leakage}} \big) + \underbrace{T \cdot (E_{\text{CMP}} + E_{\text{threshold}}^{\text{Read}})}_{\text{Thresholding}} \bigg) \quad (32)$$

Key differences between Otters and other typical SNNs include replacing digital weight reads with lower-energy analog reads from the TFT ($E_{\text{Analog}}^{\text{Read}}$) and, for the QKV calculation, using an energy-efficient binary KV read ($E_{\text{binarykv}}^{\text{Read}}$). The energy model is configured for a BERT-base architecture with a batch size ($B$) of 64, a sequence length ($S$) of 128, and input/output channel dimensions ($C_i, C_o$) of 768. The model features 12 attention heads ($h$), with a per-head dimension ($d_k$) of 64. Energy costs are derived from established models. For FP32 operations, we assume that a multiply-accumulate (MAC) consumes 4.6 pJ and a clamp operation consumes 0.9 pJ which is from (Horowitz, 2014) cause our platform doesn't support FP calculations. For our INT4 models, we differentiate between a 4-4-16bits MAC (0.0848 pJ) and a 1-4-16bits MAC (0.0663 pJ). The costs for 4-16-16bits, 2-16-16bits and 1-16-16bits ACC are 0.0502 pJ, 0.0477 pJ and 0.0429 pJ, respectively. SNN-specific operations, such as comparison and subtraction, are each modeled at 0.0502 pJ. The total energy for an analog read operation is 0.0246 pJ (0.00875pJ for power the TFT, 1.33e-6pJ for sampling, 0.0053pJ for ADC including the amplifier and 0.010505 for the 4bits LUT). The model accounts for a static leakage energy $E_{\text{leakage}}$ of 0.002 pJ per cycle, a weight activation (read/write) cost of 0.0985 pJ/bit, and a sparse data movement cost of 0.18 pJ per bit. All integer compute, data movement, and memory access costs are based on measurements from a commercial 22nm process, while the floating-point compute energy (Horowitz, 2014) and ADC values (Su et al., 2023) are taken from existing literature.

### A.3.2 DETAILED ENERGY BREAKDOWN

Table 5 and Figure 6 detail the energy breakdown of Otters-1bitkv. Spike movement is the main energy consumer, accounting for approximately 55% of the total. The next largest contributor is membrane potential accumulation (Mem. ACC), while K/V read/write (K/V R/W) consumes the least energy.

Table 5: Energy breakdown of the Otters self-attention block (in mJ). "Mem. ACC" denotes membrane-potential accumulation; "K/V R/W" denotes key/value read/write.

| Block | Compute | | Data movement | | Read analog | Leakage | Total |
|---|---|---|---|---|---|---|---|
| | Mem. ACC | Thresh. | Spike movement | K/V R/W | | | |
| Opto-FC | 0.5611 | 0.1257 | 2.0117 | 0.0012 | 0.2744 | 0.4349 | 3.4091 |
| Opto-qkv | 0.0623 | 0.1676 | 0.2235 | 0.1223 | 0.0305 | 0.0483 | 0.6546 |
| **Total** | **0.6235** | **0.2933** | **2.2352** | **0.1235** | **0.3049** | **0.4832** | **4.0637** |

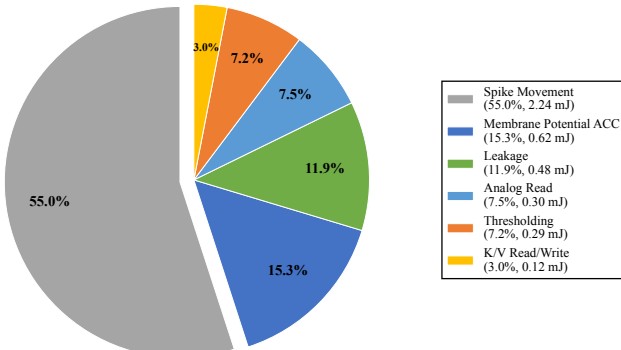

Figure 6: Proportional Energy Breakdown of Otters-1bitkv.

### A.3.3 ABLATION STUDY OF SOFTMAX AND LAYER NORMALIZATION

We initially omitted Softmax and Layer Normalization from our energy analysis, as their consumption is negligible compared to the matrix multiplication operations. In this section, we present an ablation study that accounts for these operations. We adopt the energy models from Sorbet (Tang et al., 2025), which define the costs for Softmax as $O(d)$ ACC, $O(d)$ division, and $O(d)$ exponential operations. The same model defines LayerNorm costs as $O(5d)$ ACC, $O(3d)$ MAC, $O(d)$ division, and $O(1)$ exponential operations. We calculated the total energy for one complete transformer block in Figure 7, including QKV projections, attention score calculations, the attention output projection, Softmax, Layer Normalization, and the feedforward layers (intermediate size of 3072). The results show that Softmax and LayerNorm account for only $0.32\%$ of the energy in the Otters block. Also, Otters remains the most efficient method compared to related works.

### A.3.4 ABLATION STUDY OF USING OTTERS AND TRADITIONAL TTFS METHODS

We can group TTFS encoding into two main classes: continuous-time TTFS and quantized-time TTFS. In other work, like TTFSFormer (Zhao et al., 2025), the time domain is continuous. This means they pass the time-to-first-spike between layers using floating-point (FP32) values. This high-precision FP32 number helps keep the SNN accurate after the ANN–SNN conversion. However, comparing the energy of these continuous-time methods with our approach, which uses quantized time, is not fair. In TTFSFormer, their main math unit (e.g., the accumulator, ACC) is FP32 and costs about 0.9 pJ per operation, while our INT4 ACC only costs 0.05 pJ. Also, the data movement cost grows with bit-width. This means continuous-time TTFS designs use much more memory and movement energy than low-bit designs.

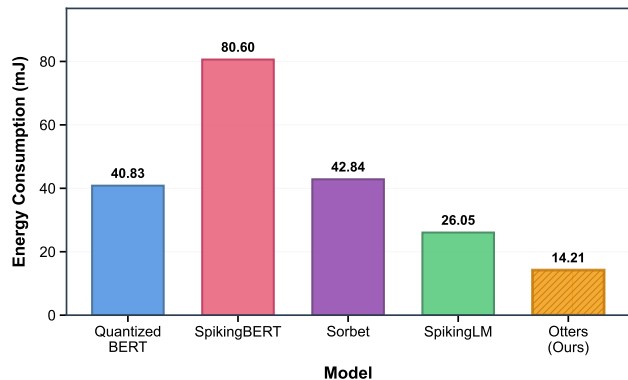

Figure 7: Energy Consumption per Transformer Block with softmax and layerNorm.

To make a fair comparison, we re-evaluated the continuous-time TTFS method in the same quantized time domain as Otters. We did this by quantizing the spike times while keeping the weights, network structure, and all other hyperparameters the same. This lets us clearly see the benefit of Otters' analog-based computing. Traditional quantized-time TTFS requires digital encoding, additional MAC operations ($E_{\text{encoding}}+E_{\text{MAC}}$), and extra weight accesses. With $T{=}15$ using the simplest $T{-}t$ encoding (4–4–4-bit ACC, $0.0163\,\text{pJ}$), traditional TTFS attention block consumes $5.12\,\text{mJ}$, $26.1\%$ more energy than 1bit Otters.

## A.4 FORWARD PROCESS OF OTTERS

The forward process is shown in Algorithm 1:

---

**Algorithm 1** Otters Neuron Forward Pass (TTFS)

---

1: **Inputs:** Presynaptic spikes $\{s_i^{l-1}(t)\}_{i=1}^{N_{in}}$; weights $\{\gamma_{ij}^l\}_{i=1}^{N_{in}}$; bias $b_j^l$; threshold function $\theta^l(t)$; total time steps $T$.
2: **Output:** Postsynaptic spike train $\{s_j^l(t)\}_{t=1}^T$.
3: $V_j \leftarrow b_j^l$ {Initialize membrane potential with bias}
4: $s_j^l[1:T] \leftarrow 0$ {Initialize output spike train to zeros}
5: has_fired $\leftarrow$ **false**
6: **for** $t = 1, 2, \ldots, T$ **do**
7:     **for** $i = 1, 2, \ldots, N_{in}$ **do**
8:         **if** $s_i^{l-1}(t) = 1$ **then**
9:             $v_i \leftarrow \gamma_{ij}^l \cdot O(t)$ {Compute PSP using the physical decay function $O(t)$}
10:             $V_j \leftarrow V_j + v_i$ {Accumulate potential}
11:         **end if**
12:     **end for**
13:     **if** $V_j \geq \theta^l(t)$ **and not** has_fired **then**
14:         $s_j^l(t) \leftarrow 1$ {Fire a spike at the current time step}
15:         has_fired $\leftarrow$ **true**
16:         **break** {TTFS constraint: emit at most one spike and stop}
17:     **end if**
18: **end for**
19: **return** $s_j^l$

---

## A.5 QNN TRAINING ALGORITHM

To mitigate the accuracy degradation from quantization, we employ a knowledge distillation (KD) framework for our quantization-aware training (Liu et al., 2022; Tang et al., 2025). This approach

Table 6: Task-specific training hyperparameters.

| Task | Max. Seq. Length | Batch Size | Learning Rate |
|------|------------------|------------|---------------|
| MNLI | 128 | 80 | $2 \times 10^{-4}$ |
| MRPC | 128 | 80 | $5 \times 10^{-5}$ |
| SST-2 | 64 | 80 | $1 \times 10^{-4}$ |
| STS-B | 128 | 80 | $5 \times 10^{-5}$ |
| QQP | 128 | 80 | $2 \times 10^{-5}$ |
| QNLI | 128 | 80 | $2 \times 10^{-4}$ |
| RTE | 128 | 80 | $5 \times 10^{-5}$ |

transfers inductive biases from a pre-trained, full-precision teacher model to the quantized student network. Our training objective is a hybrid loss function that aligns both the output distributions and the intermediate representations of the student with those of the teacher. The total distillation loss $L_{KD}$ is a weighted combination of two components: a logits distillation loss $L_{\text{logits}}$ and a representation distillation loss $L_{\text{reps}}$:

$$L_{KD} = L_{\text{logits}} + \lambda L_{\text{reps}} \tag{33}$$

where $\lambda$ is a hyperparameter that balances the two terms.

The first component, $L_{\textbf{logits}}$, is the Kullback-Leibler (KL) divergence between the teacher's soft target distribution $p$ and the student's output distribution $q$. This loss encourages the student to learn the teacher's predictions and its understanding of inter-class similarities.

$$L_{\text{logits}} = \text{KL}(p||q) \tag{34}$$

The second component, $L_{\textbf{reps}}$, minimizes the Mean Squared Error (MSE) between the intermediate feature representations from corresponding transformer blocks of the teacher ($r_i^t$) and the student ($r_i^s$). This forces the student to mimic the teacher's internal representation structure.

$$L_{\text{reps}} = \sum_i \|r_i^t - r_i^s\|_2^2 \tag{35}$$

The student model is trained end-to-end by minimizing $L_{KD}$ using the Adam optimizer, while the teacher model's weights remain frozen. We use task-specific hyperparameters for training, which are detailed in Table 6. All models are trained on A100 GPUs.

## A.6    EVALUATION BENCHMARK

We evaluated our model, Otters, on seven datasets from the GLUE benchmark:

- MNLI (Multi-Genre Natural Language Inference): A large-scale, crowdsourced collection of sentence pairs annotated for textual entailment across multiple genres.
- QQP (Quora Question Pairs): A paraphrase identification task to determine if two questions from Quora are semantically equivalent.
- QNLI (Question-Answering NLI): A natural language inference task converted from the Stanford Question Answering Dataset (SQuAD), where the goal is natural language inference
- SST-2 (Stanford Sentiment Treebank): A single-sentence classification task for sentiment analysis (positive or negative) on movie reviews.
- STS-B (Semantic Textual Similarity Benchmark): A regression task to predict the degree of similarity (on a 1–5 scale) between sentence pairs drawn from news headlines, video titles, and image captions.
- RTE (Recognizing Textual Entailment): A compilation of data from several textual entailment challenges, using text from news articles and Wikipedia.
- MRPC (Microsoft Research Paraphrase Corpus): A paraphrase detection task using sentence pairs from online news sources. The dataset is imbalanced, with 68% of pairs being paraphrases.

## A.7 INFERENCE TIME ANALYSIS

In Otters, the inference latency is mainly determined by the duration of the optical decay function (Figure 2b in the revised manuscript). Within this time window, the SNN neuron compares its accumulated membrane potential with the dynamic threshold at each time step $t_k$. If the membrane potential is larger than the threshold, the neuron fires. It then samples the value from the thin-film transistor and adds it to the next neuron's membrane potential. Once the decay period finishes, the next layer is ready to start. Thus, in theory, the computation for one layer is completed within a single time window. Taking a transformer block as an example, the calculation requires roughly 8 optical cycles: 6 cycles for the Self-Attention module (projections of Q, K, V, their multiplication and output) and 2 cycles for the Feedforward module.

In our current measurements, the transient optoelectronic response was recorded using a source-measure unit (SMU), whose minimum integration time is on the order of tens of microseconds. To ensure a window that can reliably acquire complete attenuation and fit parameters, we used a conservative sampling configuration, which resulted in a coding window of about 100 $\mu$s. This results in a total latency of 0.8 ms for one block. Further, in theory, the window can be compressed by faster readout and higher light intensity.

## A.8 ACCURACY GAP ANALYSIS

In this section, we show an analysis to separate the two sources of accuracy loss: the Quantization step (Full BERT to QNN) and the Otters implementation step (QNN to Otters). As the Table 7 shows, the main drop in accuracy occurs during the first step which is the expected performance loss from quantizing the model's weights and activations. We then measured a second, much smaller accuracy drop when converting from the QNN to our Otters SNN. The average drop is 0.59%, which mainly comes from sampling errors (the exact physical time $t_k$ needed for value $O(t_k)$ might fall between two sampling points, causing a representation error). We believe this small accuracy trade-off is highly favorable, as the Otters implementation achieves this performance while consuming only 36.8% of the QNN energy.

Table 7: Accuracy gap analysis on the GLUE benchmark, where all scores are accuracy except for STS-B (Pearson correlation).

| Model | QQP | MNLI-m | SST-2 | QNLI | RTE | MRPC | STS-B | Avg |
|---|---|---|---|---|---|---|---|---|
| Full BERT | 91.3 | 84.7 | 93.3 | 91.7 | 72.6 | 88.2 | 89.4 | 87.31 |
| QNN | 87.81 | 78.40 | 91.28 | 86.80 | 70.03 | 86.51 | 85.87 | 83.81 |
| Otters | 87.67 | 78.50 | 91.28 | 86.42 | 68.95 | 84.56 | 85.19 | 83.22 |

## A.9 GENERALIZABILITY OF OTTERS

Otters is proposed to optimize the temporal decay function and its subsequent multiplication by sampling the natural decay of an oxide optoelectronic synapse. In other words, Otters provides an analog alternative to digital matrix multiplication, which is a basic and commonly used unit in most models. Therefore, Otters has the potential to be applied to most network structures (or at least optimize parts of them) since it is an optimization for base units. In Table 8, we provide VGG/ResNet on CIFAR-10 with different time window size as an example to demonstrate the generalizability.

Table 8: Top-1 accuracy (%) of SNNs converted from ResNet18 and VGG16 across different time steps $T$. The time steps $T = 2^n - 1$ (for $n = 2, 3, 4, 5$) are selected based on the n-bit QNN conversion discussed in Proposition 1.

| Model | Time steps $T$ | | | |
|---|---|---|---|---|
| | 3 | 7 | 15 | 31 |
| Otters-ResNet18 | 89.27 | 89.31 | 92.02 | 92.05 |
| Otters-VGG16 | 87.38 | 89.42 | 91.62 | 91.49 |

## A.10   1-BIT KV ENERGY ABLATION STUDY

In this section, we recalculate the energy of other SNN methods, assuming all K/V projections are quantized to 1-bit for a fair comparison with our Otters-1bitkv model. The spike rates and time window sizes are set to be consistent with those in Appendix A.3.1. As shown in Table 9, Otters still achieves the best energy efficiency within this 1-bit KV setting.

Table 9: Energy consumption analysis on the SST-2 dataset. The Energy Ratio is Energy(Otters) / Energy(Model).

| Model | FC (mJ) | QKV (mJ) | Total (mJ) | Energy Ratio |
|---|---|---|---|---|
| Sorbet (1bit kv) (Tang et al., 2025) | 3.39 | 0.58 | 11.32 | 2.79x |
| SpikingBERT (1bit kv) (Bal & Sengupta, 2024) | 6.37 | 1.07 | 21.27 | 5.24x |
| SpikingLM (1bit kv) (Xing et al., 2024b) | 2.09 | 0.35 | 6.97 | 1.72x |
| **Otters (1bit kv)** | **1.14** | **0.33** | **4.06** | **1x** |

## A.11   QNN-TO-SNN CONVERSION DISCUSSION

Direct training of SNNs is challenging because the binary spike activation is non-differentiable. Instead of using approximate gradients, QNN-to-SNN conversion trains a functionally equivalent QNN first and transfers the weights to the SNN. This approach often yields higher accuracy for deep networks. Many related works have implemented this. For example, in language tasks, Sorbet achieves 79.80% accuracy on GLUE using an average spike generation method (Tang et al., 2025); for vision tasks, Wang et al. reach 85.31% on ImageNet using a Multi-basis Exponential Decay (MBE) neuron (Wang et al., 2025a). Liu et al. (Liu & Liu, 2018) propose quantization-aware DNNs using neuron convergence and weight clustering, and deploy the resulting 4-bit networks on memristor-based Spiking neuromorphic computing systems, achieving 90.33% for CIFAR-10. Beyond these, Ajay et al. (Ajay et al., 2024) introduce MC-QDSNN, a quantized deep evolutionary SNN that uses Multi-Compartment Leaky (MCLeaky) neurons for better and more efficient modeling of time-series data. However, most prior works rely on rate encoding and do not target optical hardware. In contrast, Otters focuses on TTFS-based SNNs using optical spiking neurons. To adapt QNNs to our $In_2O_3$ devices, we propose specific conversion techniques, including dynamic thresholding and mapping quantization bits to the natural decay. This allows us to replace expensive digital matrix multiplication with energy-efficient analog sampling.

## A.12   THE USE OF LARGE LANGUAGE MODELS

We use LLMs to aid or polish writing.

