# OpenReview forum: "Otters: An Energy-Efficient Spiking Transformer via Optical Time-to-First-Spike Encoding"
_ICLR.cc/2026/Conference — ICLR 2026 Poster_

### Official Review · Reviewer_YQwJ · 2025-10-31

**Soundness:** 3
**Presentation:** 3
**Contribution:** 3
**Rating:** 6
**Confidence:** 2

**Summary:**

The paper presents an interesting hardware-software co-design for ultra-low-energy Spiking Neural Networks (SNNs). The core innovation is the OTTERS optoelectronic synapse, which repurposes the natural signal decay of a custom-fabricated Indium Oxide (In2O3) thin-film transistor (TFT) to physically implement the temporal decay function required by Time-to-First-Spike (TTFS) encoding. This strategy eliminates the costly digital computation that has plagued traditional TTFS approaches.

**Strengths:**

Repurposing the natural decay of a fabricated In2O3 optoelectronic TFT to implement TTFS temporal decay is original and elegant, it turns a device “bug” into a computational primitive.

Practical engineering choices are well motivated to avoid run-time multiplications and to exploit sparsity in TTFS.

The work is novel, impactful, and supported by extensive experiments and a rigorous analytical model.

**Weaknesses:**

The energy numbers rely on an analytical model mixing measured device values (ADC, TFT) with literature energy numbers for ACC/NoC/etc. However, the headline claims (Otters 4.06 mJ per attention block, etc.) would require clearer provenance.

Table 1 and Table 2 compare to SNN and QNN baselines, but it is unclear whether baselines were re-run (or taken from the original work) with identical data preprocessing, same batch length/sequence length, or whether SOP/spike rates were taken from original papers.

**Questions:**

In the energy estimations, which terms dominate (analog read, ADC, spike movement, threshold reads, K/V write)? Provide a breakdown per term with uncertainties and explicitly list all constants used and their sources.

In the baseline comparisons, clarify experimental protocol and, where feasible, re-evaluate key baselines under the same measurement assumptions or provide scripts to reproduce the energy comparison.

Minor:
- In Table 2, for the last column Energy Ratio, I suggest using an upward arrow (or statement in the caption) to clearly indicate that the higher is better.

- Please add the date for the ICML paper of Xing et al. line 588 (2024), and clarify which paper by Xingrun Xing is referred to by "Xing et al." since you cite two papers in the same year with same first author.

---

> ### Author Response · Authors · 2025-11-20
> **Response to Reviewer YQwJ**
>
> **Response for W1/Q1**: Thank you for this great suggestion. Adding an energy breakdown analysis improves the clarity of our energy results. We have added both a table and a pie chart to show the detailed breakdown of energy consumption. We analyzed these new results in Appendix A.3.2 (highlighted in blue). As the table shows, data movement, especially spike movement, dominates the energy consumption.
>
>
> **Table 1: Energy breakdown of the Otters self-attention blocks (in mJ).** "Mem. ACC" denotes membrane-potential accumulation; "K/V R/W" denotes key/value read/write.
>
> | Block | Comp. (Mem. ACC) | Comp. (Thresh.) | Data movement  (Spike mov.) | Data movement (K/V R/W) | Read analog | Leakage | Total |
> | :--- | :---: | :---: | :---: | :---: | :---: | :---: | :---: |
> | Opto-FC | 0.5611 | 0.1257 | 2.0117 | 0.0012 | 0.2744 | 0.4349 | 3.4091 |
> | Opto-qkv | 0.0623 | 0.1676 | 0.2235 | 0.1223 | 0.0305 | 0.0483 | 0.6546 |
> | **Total** | **0.6235** | **0.2933** | **2.2352** | **0.1235** | **0.3049** | **0.4832** | **4.0637** |
>
> In Section A.3.1, we improved the clarity by listing the constants used. These include the parameters for a BERT-base architecture: a batch size ($B$) of 64, a sequence length ($S$) of 128, input/output channel dimensions ($C_i, C_o$) of 768, and 12 attention heads ($h$) with a per-head dimension ($d_k$) of 64. Regarding the energy sources, we measured the TFT energy by integrating the instantaneous power ($P = V \cdot I$) over the duration of each operation. All integer compute, data movement, and memory access costs are based on measurements from a commercial 22nm process, while the floating-point compute energy and ADC values are taken from existing literature. We have improved the writing for all energy numbers to clearly explain and cite their sources.
>
> ---
>
> **Response for W2/Q2**: Thank you for pointing this out. To improve the clarity of our paper, we updated Appendix A.3.1 to explain the experimental protocol and measurement assumptions explicitly.
>
> For the accuracy results in Table 1, we used the values reported directly in the original papers. We assume these papers represent the best possible performance for those methods. If Otters achieves higher accuracy, it demonstrates that our method is sufficiently competitive.
>
> However, for the energy calculation in Table 2, we did not use the energy numbers reported in the original papers. Energy depends heavily on batch size and sequence length, so a direct comparison is often unfair. Also, most related works do not consider hardware overhead, which actually dominates the total energy (as shown in our previous response). Therefore, we recalculated their energy using the energy equations we built for them, combined with the spike rates and time window sizes reported in their papers. This can also ensure a fair comparison under the same assumptions (model size, batch size, and sequence length). We have provided the reproduction scripts in the “energy_cal" folder of the Supplementary Material.
>
> ---
> **Response for minor questions**: Thank you for your suggestion. We have added the upward arrow and fixed the citation problems.

---

### Official Review · Reviewer_AM8c · 2025-11-01

**Soundness:** 3
**Presentation:** 4
**Contribution:** 2
**Rating:** 6
**Confidence:** 3

**Summary:**

This submission implements spiking neural networks (SNNs) on optoelectronic devices to mitigate the high computational cost associated with evaluating temporal decay functions and their subsequent multiplications with synaptic weights. Specifically, the authors fabricate a custom indium oxide–based optoelectronic synapse capable of directly realizing the required temporal function. Furthermore, they introduce a quantized neural network–to–SNN conversion algorithm to enhance the efficiency and practicality of the proposed system.

**Strengths:**

The work elegantly bridges neuromorphic hardware and algorithmic innovation by exploiting the intrinsic physical decay of optoelectronic devices to implement TTFS encoding. This approach offers a conceptually novel solution to the costly digital decay function in TTFS.

The authors provide a rigorous and realistic energy model covering compute, data movement, and analog operation costs, grounded in commercial 22nm process measurements.

The paper contributes a tangible step toward neuromorphic computation for language models, merging physical device dynamics with algorithmic efficiency, an area of high impact given the growing energy concerns of LLMs.

**Weaknesses:**

**i**. While the concept is elegant, it remains unclear how scalable the proposed In₂O₃ optoelectronic synapses are for large-scale deployment or integration into full neuromorphic hardware systems. For instance, what is the device size of otters for BERT?

**ii**. The transition from quantized neural networks (QNNs) to spiking neural networks (SNNs) is not a new research question, as several prior works have explored QNN-to-SNN conversion frameworks. However, this submission does not provide a sufficient comparative analysis with existing methods, like [1][2]. A more detailed discussion is needed to highlight the unique technical contributions, methodological distinctions, and advantages of the proposed conversion algorithm over established approaches.

**iii**. The experiments are conducted only using BERT as the base architecture. It remains unclear whether the proposed Otters framework can generalize to other Transformer variants or model families.

**iv**. The reported energy savings appear to stem primarily from quantization effects rather than the proposed optoelectronic TTFS mechanism itself. In particular, the ablation study compares a 4-bit SNN with a 1-bit Otters, which inherently favors the latter due to lower bit precision. This comparison seems unfair and may overstate the true contribution of the proposed hardware-software co-design. A more controlled comparison at equal bit-widths would provide a clearer and more convincing evaluation of Otters’ energy efficiency.

[1] Liu, Fuqiang, and Chenchen Liu. "Towards accurate and high-speed spiking neuromorphic systems with data quantization-aware deep networks." Proceedings of the 55th Annual Design Automation Conference. 2018.

[2] Ajay, B. S., and Madhav Rao. "MC-QDSNN: Quantized Deep evolutionary SNN with Multi-Dendritic Compartment Neurons for Stress Detection using Physiological Signals." IEEE Transactions on Computer-Aided Design of Integrated Circuits and Systems (2024).

**Questions:**

1. Since the experiments are conducted through simulation rather than on real optoelectronic hardware, it remains unclear how much of the performance degradation shown in Table 1 originates from the Otters implementation itself, as opposed to the quantization step during the QNN-to-SNN conversion. A clearer analysis quantifying the accuracy gap relative to standard BERT, after isolating the impact of quantization, would help clarify the true effect of the proposed Otters mechanism on model accuracy.


2. Can you provide the reference to the cost computation functions in A 3.1. Without proper references, these cost formulations appear insufficiently justified, as they seem to assume idealized conditions rather than reflecting the complexities of realistic hardware environments.

---

> ### Author Response · Authors · 2025-11-20
> **Response to Reviewer AM8c (1/2)**
>
> **Response to Q1**: This is a great suggestion that improves the clarity of our paper. We thank the reviewer for pointing this out. In Section A.8 of the revised manuscript, we have added an analysis to separate the two sources of accuracy loss: the quantization step (full precision BERT to QNN) and the Otters implementation step (QNN to Otters). As the Table shows, the main drop in accuracy occurs during the first step of quantization, which is expected. We then measured the second, much smaller accuracy drop when converting from the QNN to our Otters SNN. The average drop is 0.59\%, mainly due to sampling errors (the exact physical time $t_k$ required for the value $O(t_k)$ might fall between two sampling points, leading to a representation error). We believe this small accuracy trade-off is acceptable given that Otters consumes only 36.8\% of the QNN's energy.
>
> **Table 2: Accuracy gap analysis on GLUE benchmark.**
>
> | Model | QQP | MNLI-m | SST-2 | QNLI | RTE | MRPC | STS-B | Avg |
> | :--- | :---: | :---: | :---: | :---: | :---: | :---: | :---: | :---: |
> | Full BERT | 91.3 | 84.7 | 93.3 | 91.7 | 72.6 | 88.2 | 89.4 | 87.31 |
> | QNN | 87.81 | 78.40 | 91.28 | 86.80 | 70.03 | 86.51 | 85.87 | 83.81 |
> | Otters | 87.67 | 78.50 | 91.28 | 86.42 | 68.95 | 84.56 | 85.19 | 83.22 |
>
> ---
> **Response to Q2**: Thank you for the suggestion. We have included references [9, 10] in Section 3.4, which provide the basis for our cost functions and clarify the energy equations. We acknowledge that evaluating energy efficiency is complex as it is heavily dependent on model design, software mapping, and hardware specifics. To construct our analytical model, we followed the methodology and conditions in [9]. We also note that many prior SNN studies focus mainly on compute energy. We tried to provide a more complete assessment by also accounting for hardware costs, including data movement, leakage, and analog energy. We believe this approach better reflects energy consumption. We have updated the revised manuscript to cite these references and discuss the settings in detail.
>
> [9] Yan, Zhanglu, Zhenyu Bai, and Weng-Fai Wong. “Reconsidering the energy efficiency of spiking neural networks." arXiv preprint arXiv:2409.08290 (2024).
>
> [10] Dampfhoffer, Manon, et al. “Are SNNs really more energy-efficient than ANNs? An in-depth hardware-aware study." IEEE Transactions on Emerging Topics in Computational Intelligence 7.3 (2022): 731-741.
>
> ---
> **Response to W1**:
> We thank the reviewer for this question. We have added a new section (Section 5) in the revised manuscript to discuss this.
> Our current In$_2$O$_3$ synapse is about 0.012 mm$^2$ with 30 $\mu$m channel length. For large-scale deployment, we use a device reuse strategy. Since we quantize weights to 1-bit (Section 3.3), the decay function is identical for the entire layer. This allows us to share just one physical device per layer in the best case. Therefore, one BERT block (Self-Attention + Feedforward) would need roughly eight devices. Thus, the total optical area for one block is ~0.096 mm$^2$. Furthermore, the 0.012 mm$^2$ size is limited by the precision of our lab fabrication equipment, not the technology itself. Recent studies show these devices can scale down to 40 nm [11] and can be arranged in 3D stacked layers, which would significantly reduce the area. In the same newly added section of the revised manuscript, we discuss the possibility of improving the size.
>
> Regarding integration, recent studies show that In$_2$O$_3$ devices are compatible with 3D stacking on CMOS chips. This is because they have a low thermal budget ($\le$ 300 °C) and good uniformity [12]. This provides a practical path for vertical integration. However, challenges remain, such as managing wire density across stacked layers and ensuring accurate layer alignment. We also discuss this in the new section to make our paper more complete.
>
>
> [11] Si, Mengwei, et al. “High-performance atomic-layer-deposited indium oxide 3-D transistors and integrated circuits for monolithic 3-D integration." IEEE Transactions on Electron Devices 68.12 (2021): 6605-6609.
>
> [12] Tang, W., et al. “Monolithic 3D integration of vertically stacked CMOS devices and circuits with high-mobility atomic-layer-deposited In 2 O 3 n-FET and polycrystalline Si p-FET: Achieving large noise margin and high voltage gain of 134 V/V." 2022 International Electron Devices Meeting (IEDM). IEEE, 2022.
>
> ---
> **Response to W2**:  We thank the reviewer for this important comment. We have added a detailed discussion in Appendix A.11 to compare existing QNN-to-SNN methods. We discuss their methods, applications, and limitations, and how they differ from our proposed method. This should make our contributions clearer.

---

> > ### Author Response · Authors · 2025-11-20
> > **Response to Reviewer AM8c (2/2)**
> >
> > **Response to W3**:  We thank the reviewer for this concern regarding generalizability. Otters is designed to optimize the temporal decay function and its subsequent multiplication by sampling the natural decay of an oxide optoelectronic synapse. In other words, Otters provides an analog alternative to digital matrix multiplication. Since matrix multiplication is a fundamental operation in most neural networks, Otters can be applied to other Transformer variants and other model families, or at least improve the computational efficiency of specific parts of them. We have added a discussion on generalizability in Appendix A.9.
> >
> > ---
> > **Response to W4**: We thank the reviewer for this concern regarding the fairness of the comparison. First, we want to clarify that the SNNs compared in Table 2 also use 1-bit weights. We have updated the text to make this clear and prevent misunderstanding.
> >
> > To further address the concern, we added a detailed ablation study in Appendix A.10. Here, we compare Otters (1-bit weight, 1-bit KV) against other SNNs with the same 1-bit settings. Additionally, we compared Otters to a QNN with identical bit-width (1-bit weight, 1-bit KV) and network structure. The results show that converting this QNN to Otters reduces the energy from 11.03 mJ to 4.06 mJ. The revised manuscript now includes controlled comparisons at equal bit-widths for QNNs, related SNNs, and Otters. In all cases, Otters demonstrates the best energy efficiency.

---

> > > ### Comment · Reviewer_AM8c · 2025-11-22
> > >
> > > Thanks for your sincere response that has addressed my concerns. I would like to raise my score from 6 to 8. Good luck.

---

> > > > ### Author Response · Authors · 2025-11-22
> > > > **Response to Reviewer AM8c**
> > > >
> > > > We sincerely thank you for taking the time to review our responses and for updating the review!

---

### Official Review · Reviewer_Ei9k · 2025-11-04

**Soundness:** 3
**Presentation:** 3
**Contribution:** 3
**Rating:** 8
**Confidence:** 4

**Summary:**

This paper proposes to use opto-electrical neurons for low-power SNN acceleration using time to first spike encoding that naturally aligns with the opto electrical neuron's characteristics.  The energy analysis and accuracy analysis is quite reasonable.  The conversion approach and quantization techniques as well as the noise-aware training all appear well explained and reasonable.

**Strengths:**

I think the novelty of the approach and the write-up and analysis are solid. Well explained paper and concept.  I was very happy to see the energy analysis use estimates from Horowitz which are realistic and capture the cost of ADCs and data movement.

**Weaknesses:**

It seems the opt-electrical device is not small compared to a digital SNN accumulator.  It seems to me that the scale of the hardware may be the biggest challenge in making such an approach practical and yet there seems to be no description of this challenge in the paper, even in the future work.

I think even for a conference like ICLR and perhaps especially for a conference like ICLR where the audience may not realize the fundamental difference in hardware scale, this should be mentioned and scaling efforts for this technology should be at least outlined.  Otherwise, this is yet another pipedream in the field of energy-efficient SNN hardware. More specifically, how many such devices might be needed to have the same order of magnitude in compute latency and what, with current technology, would the size of such a chip be.  What are the challenges of heterogeneously integrating such devices with SOTA digital hardware?

**Questions:**

Please see weaknesses.

I also heard of other SOTA SNNs such as QKFormer. Can you comment on how this work relates to yours? See https://arxiv.org/abs/2403.16552.

**Details Of Ethics Concerns:**

None.

---

> ### Author Response · Authors · 2025-11-20
> **Response to Reviewer Ei9k (1/2)**
>
> **Response to the weakness about hardware scale**: We thank the reviewer for this concern regarding the hardware scale and practicality. We fully agree that the device footprint is a critical factor in determining whether this optoelectronic SNN approach can become viable. We have added a new section (Section 5) in the revised manuscript to discuss the hardware scale problem and its future works.
>
> The In$_2$O$_3$ optoelectronic synapse in our prototype has an effective area of about 0.012 mm$^2$ with a 30 $\mu$m channel length.  The precision of our fabrication equipment mainly limits this size, and it could be much smaller. Recent work has demonstrated In$_2$O$_3$ transistors with channel lengths down to 40 nm [2], showing that nearly three orders of magnitude can reduce the device area. In addition, several studies have shown that oxide devices can be integrated into 3D-stacked layers, providing another path further to reduce the effective area per synapse [3,4,5]. With the TTFS mechanism verified, we plan to focus on device scaling in our future work.
>
>
> We also wish to point out that the device count can be reduced by device sharing. For example, consider a Q-projection layer with input channels $C_i$ and output channels $C_o$. If every weight required one device, the layer would need $O(C_i \times C_o)$ devices. In the Otters setting, this number would reach $10^6$–$10^7$, which is clearly not practical. However, in our setting (introduced in Section 3.3), all weights are quantized to 1-bit. This means the decay function is identical across the entire layer, allowing the In$_2$O$_3$ device to be shared. In the best case, one FC layer can reuse just one physical device. Thus, a single BERT block would require about 8 devices, leading to ~0.096mm$^2$ areas: 6 for the self-attention module (projections of Q, K, V, their multiplication and output) and 2 for the feed-forward module. (In real systems, the number of devices might be higher, and sharing may not be ideal due to architecture and compiler limits.) The next step in our work is to design an architecture that increases device reuse to reduce chip area further. We have included this discussion in the new section.
>
> Regarding integration with state-of-the-art digital hardware, recent studies show that In$_2$O$_3$ TFTs are compatible with 3D stacking on CMOS chips. This is because they have a low thermal budget ($\leq$ 300 °C) and good uniformity [2]. This provides a practical path for vertical integration. However, several challenges remain, such as managing wire density in stacked layers, aligning the photodetectors/TFT layers with metal layers, and handling device variability in large arrays.
>
>
> [2] Si, Mengwei, et al. “High-performance atomic-layer-deposited indium oxide 3-D transistors and integrated circuits for monolithic 3-D integration." IEEE Transactions on Electron Devices 68.12 (2021): 6605-6609.
>
> [3] Tang, W., et al. “Monolithic 3D integration of vertically stacked CMOS devices and circuits with high-mobility atomic-layer-deposited In 2 O 3 n-FET and polycrystalline Si p-FET: Achieving large noise margin and high voltage gain of 134 V/V." 2022 International Electron Devices Meeting (IEDM). IEEE, 2022.
>
> [4] Yuvaraja, Saravanan, et al. “Three-dimensional integrated metal-oxide transistors." Nature Electronics 7.9 (2024): 768-776.
>
> [5] Kwak, Jungyoun, et al. “Monolithic 3D transposable 3T embedded DRAM with back-end-of-line oxide channel transistor." 2024 IEEE International Symposium on Circuits and Systems (ISCAS). IEEE, 2024.

---

> > ### Author Response · Authors · 2025-11-20
> > **Response to Reviewer Ei9k (2/2)**
> >
> > ***Response to the question about related works***: Thank you for this question and for bringing the QKFormer paper to our attention. To us, QKFormer and Otters both aim to improve energy efficiency in spiking neural networks, but the approaches are completely different. QKFormer represents a group of works that focus on algorithmic and architectural improvements [6,7,8]. It optimizes the attention mechanism from $O(N^2)$ to linear complexity (linear-attention SNNs). Otters, on the other hand, focuses on hardware-software co-design to optimize a core computational operation. For example, we replace the temporal decay function and multiplication by sampling the natural decay of an oxide optoelectronic synapse. This shifts the computation from digital to analog.
> >
> > The two directions are complementary and address different aspects of the problem. We have added a discussion of the relationship between our work and linear-attention SNNs, such as QKFormer, to Section 5 of the revised manuscript to enrich the paper's context and highlight these complementary research directions.
> >
> > [6] Zhou, Chenlin, et al. “Qkformer: Hierarchical spiking transformer using qk attention." Advances in Neural Information Processing Systems 37 (2024): 13074-13098.
> >
> > [7] Wang, Shuai, et al. “Spiking vision transformer with saccadic attention." arXiv preprint arXiv:2502.12677 (2025).
> >
> > [8] Guo, Yufei, et al. “Spiking Transformer: Introducing Accurate Addition-Only Spiking Self-Attention for Transformer." Proceedings of the Computer Vision and Pattern Recognition Conference. 2025.

---

### Official Review · Reviewer_CymU · 2025-11-05

**Soundness:** 2
**Presentation:** 1
**Contribution:** 2
**Rating:** 4
**Confidence:** 4

**Summary:**

This paper introduces a hardware–software co-design framework named Otters, which leverages the natural signal decay in indium oxide optoelectronic transistors to physically implement TTFS computation. By interpreting the analog output as a fused representation of synaptic weight and temporal decay, the system eliminates costly digital multiplications. Furthermore, the authors propose a QNN-to-SNN conversion algorithm, integrating it into Transformer architectures. Experimental results on the GLUE benchmark demonstrate SOTA accuracy and a 1.77× improvement in energy efficiency compared to previous SNNs, validated through comprehensive measurements on a 22nm commercial process.

**Strengths:**

1. This paper presents a novel physical computing paradigm that leverages the inherent signal decay of optoelectronic devices as the temporal decay mechanism for TTFS computation, thereby eliminating the need for energy-intensive digital multiplications.

2. It introduces a theoretically grounded QNN-to-SNN conversion framework, establishing the functional equivalence between quantized neural representations and spike-based computation, which is significant for bridging conventional deep learning and neuromorphic systems.

3. The proposed Otters architecture, validated through a hardware–software co-design, achieves state-of-the-art performance on GLUE and demonstrates energy efficiency advantages.

**Weaknesses:**

1. The experimental evaluation is limited to Transformer architectures, which are not the primary application domain for SNNs. Since SNNs are particularly suited for edge computing and low-power vision tasks, extending the analysis or providing validation on CNN-based models [1] would substantially strengthen the work.

2. The proposed QNN–SNN mapping depends on a pre-trained quantized model, which introduces additional training overhead. The authors should clarify whether this preprocessing step yields tangible performance benefits and discuss whether the Otters framework provides practical deployment advantages over conventional QNNs on real hardware.

3. The experimental comparison is limited. Including results against other TTFS-based conversion methods [2] would better substantiate the claimed efficiency gains.

4. The mathematical expressions lack consistency in formatting and punctuation. All equations should be standardized, properly numbered, and punctuated to ensure clarity and formal correctness.

[1] Deep residual learning for image recognition. CVPR 2016.

[2] TTFSFormer: A TTFS-based Lossless Conversion of Spiking Transformer. ICML 2025.

**Questions:**

1. The reported energy analysis omits Softmax operation and Layer Normalization [3], both essential operations in Transformer computation. Incorporating these operations would provide a more comprehensive and realistic assessment of the total energy consumption.

2. Since the paper emphasizes energy efficiency, providing inference latency results is necessary to validate the method’s real-time performance and practicality.

3. It remains unclear whether the proposed Otters-based TTFS framework generalizes to other datasets or CNN-based architectures. Demonstrating such experiments would enhance the paper’s impact and generality.

[3] Spatio-temporal approximation: A training-free snn conversion for transformers. ICLR 2025.

---

> ### Author Response · Authors · 2025-11-20
> **Response to Reviewer CymU (1/2)**
>
> **Response to Q1**: Thank you for this suggestion. We initially omitted the energy for Softmax and Layer Normalization because their consumption is very small compared to the large matrix operations. However, we agree that including them provides a more realistic assessment. We have now added this analysis to Section A.3.3 (highlighted in blue) and reported the total energy consumption. The results confirm that Softmax and Layer Normalization account for only 0.32\% of the total energy, and Otters remains the most efficient method.
>
> ---
> **Response to Q2**: We thank the reviewer for this question regarding latency. In Otters, the inference latency is mainly determined by the duration of the optical decay function (Figure 2b in the revised manuscript). In our current measurements, we used a source-measure unit (SMU) to record the data and observed an optical decay duration of 100 $\mu$s.
>
> Since the digital operations (comparison and accumulation) are much faster than the optical decay (which is the bottleneck), they can be completed within this time window. During this decay period, at each time step $t_k$, the SNN neuron compares its membrane potential with the threshold. If it fires, it samples the current decay value from the device and adds it to the next layer's potential. Therefore, when the 100 $\mu$s decay is over, the calculation for that layer is also completed, and the accumulated membrane potential is ready for the next layer.
>
> Thus, in theory, the computation for one layer can be completed within a single time window. Taking an transformer block as an example, the calculation requires roughly 8 optical time periods: six for the self-attention module (projections of Q, K, V, their multiplication, and output) and two for the feed-forward module. This results in a total latency of ~0.8 ms for one block. We have added a discussion in Appendix A.7 to analyze the inference and potential ways to improve it in the future.
>
> ---
> **Response to Q3/W1**:  These are good suggestions. The optimization of Otters' digital matrix multiplication is also a key component of most CNN structures. To further demonstrate the generalizability of Otters, we have added experiments using VGG/ResNet on CIFAR-10 with different time window sizes. A new section, A.9, has been added (in blue in the revised manuscript), and the results show that Otters can also be applied to CNN-based models.
>
> ---
> **Response to W2**: We agree that the pre-trained quantized model introduces some training overhead. However, compared to training an SNN directly, training a similar QNN is much easier. This approach avoids the over-sparsity problem observed in TTFS SNN training and enables us to leverage existing quantized training techniques, such as knowledge distillation. We also detail our training hyperparameters in Appendix A.5 and show the training process.
>
> This preprocessing step results in significant performance benefits. Without QNN pre-training, converting an unquantized model directly to Otters results in a significant loss in accuracy. For example, in the SST-2 setting, Otters would only achieve an accuracy of 51.15\%. After QNN pre-training and conversion, it achieves 91.28\% (an improvement of over 40\%).
>
> We also show the practical advantages of conventional QNNs in deployment. The energy benefit we showed are based on a commercial 22nm process, and includes compute, data movement, memory access, and leakage energy. Compared to the pre-trained QNN (with the exact same network structure and quantization level), Otters consumes only 36.8\% of the energy. We show the detailed results in Table 2 in the revised manuscript.

---

> > ### Author Response · Authors · 2025-11-20
> > **Response to Reviewer CymU (2/2)**
> >
> > **Response to W3**: Thank you for the suggestion. We have updated Appendix A.3.4 to more clearly compare Otters with other TTFS methods.
> >
> > We can group TTFS encoding into two main classes: continuous-time TTFS and quantized-time TTFS. TTFSFormer[1] uses a continuous time domain. This is equivalent to passing the time-to-first-spike between layers as floating-point (FP32) values. This high-precision FP32 format helps maintain SNN accuracy after the ANN–SNN conversion. However, comparing the energy of these continuous-time methods with that of our approach, which uses quantized time, is unfair. In TTFSFormer, their central math unit (e.g., the accumulator, ACC) is FP32 and consumes about 0.9 pJ per operation, whereas our INT4 ACC consumes only ~0.05 pJ. Also, the cost of data movement grows with each increase in bitwidth. This means continuous-time TTFS designs use much more memory and movement energy than low-bit designs.
> >
> > To make a fair comparison, we re-evaluated the continuous-time TTFS method in the same quantized time domain as Otters. We quantized the spike times while keeping the weights, network structure, and all other hyperparameters the same. This clearly shows the benefit of Otters' analog-based computing. With $T=15$ and the simplest $T-t$ decay function encoding (using a 4-4-4-bit ACC at 0.0163 pJ), the traditional TTFS attention block uses 5.12 mJ, which is 26.1\% more energy than our 1-bit Otters design.
> >
> > [1] Zhao, Lusen, et al. “TTFSFormer: A TTFS-based Lossless Conversion of Spiking Transformer." Forty-second International Conference on Machine Learning.
> >
> > ---
> > **Response to W4**:  We apologize for this, as the math was written over an extended period of time with many changes. We have standardized the formatting and punctuation for the mathematical expressions.

---

### Official Review · Reviewer_8Ccw · 2025-11-06

**Soundness:** 2
**Presentation:** 3
**Contribution:** 3
**Rating:** 6
**Confidence:** 3

**Summary:**

This paper propose to transform the intrinsic “signal decay” of optoelectronic synapses into a computational resource, thereby achieving TTFS coding in SNNs with extremely low energy consumption. Specifically, it uses the physical decay curve of customized indium oxide transistors to replace digital computation, and then proposes a novel QNN-to-SNN conversion algorithm that maps a trained quantized Transformer into an equivalent spiking network. Finally, this paper adds 1-bit KV projection to further reduce energy consumption.

**Strengths:**

1. It proposes a new “physical computation” paradigm that achieves in-memory computation by integrating synaptic weight multiplication and temporal decay into a single optoelectronic physical process, thereby improving energy efficiency from the ground up.

2. It achieves the highest accuracy among SNNs on the GLUE benchmark and demonstrates a 1.77x energy-efficiency improvement over the previous SOTA model, which validate the effectiveness of the proposed framework.

**Weaknesses:**

1. Since the paper’s contributions involve algorithm–hardware co-design, it should provide a more detailed breakdown of energy consumption to clarify the contribution of each component, such as $(E_{\text{analog}})$.

2. The results are presented only for the BERT-Base model, which raises concerns about the generalizability of the proposed method.

3. For a fairer comparison, the paper should include both performance and energy consumption results of other models with 1-bit KV to more clearly demonstrate the energy savings achieved by the new hardware paradigm.

**Questions:**

check the weakness

---

> ### Author Response · Authors · 2025-11-20
> **Response to Reviewer 8Ccw**
>
> **Response to W1**: Thank you for your suggestion. We added both a table and a pie chart to show the detailed breakdown of energy consumption. We added and analyzed these new results (highlighted in blue) in Appendix A.3.2.
>
>
> **Table 1: Energy breakdown of the Otters self-attention blocks (in mJ).** "Mem. ACC" denotes membrane-potential accumulation; "K/V R/W" denotes key/value read/write.
>
> | Block | Comp. (Mem. ACC) | Comp. (Thresh.) | Data movement  (Spike mov.) | Data movement (K/V R/W) | Read analog | Leakage | Total |
> | :--- | :---: | :---: | :---: | :---: | :---: | :---: | :---: |
> | Opto-FC | 0.5611 | 0.1257 | 2.0117 | 0.0012 | 0.2744 | 0.4349 | 3.4091 |
> | Opto-qkv | 0.0623 | 0.1676 | 0.2235 | 0.1223 | 0.0305 | 0.0483 | 0.6546 |
> | **Total** | **0.6235** | **0.2933** | **2.2352** | **0.1235** | **0.3049** | **0.4832** | **4.0637** |
>
> ---
>
> **Response to W2**: We thank the reviewer for this concern. Otters is proposed to optimize the temporal decay function and its subsequent multiplication by sampling the natural decay of an oxide optoelectronic synapse. In other words, Otters provides an analog alternative to digital matrix multiplication, which is a basic and commonly used unit in most models. Therefore, Otters has the potential to be applied to most network structures (or at least optimize parts of them) since it is an optimization for base units. We have added a discussion on generalizability in Appendix A.9. Further, in the same section, we have also added results showing the implementation of Otters on convolutional-type networks to further demonstrate its generalizability.
>
> ---
>
> **Response to W3**: We agree with this concern about a fair comparison. To address this, we first updated the main energy comparison (Table 2) to include the energy of Otters for both 1-bit KV and 4-bit KV (rather than listing only the 1-bit KV version) for a fairer comparison. Additionally, we added a new comparison in the Appendix A.10 where we assume the other related works also quantize their KV to 1-bit as suggested. The results show that Otters still has the best energy consumption in this 1-bit KV setting. Furthermore, the accuracy of Otters (1-bit KV) is similar to, or even higher than, the other methods that do not use 1-bit KV.

---

### Author Response · Authors · 2025-12-02
**Summary of Rebuttal and Revisions for Area Chair**

We sincerely thank the reviewers for their constructive feedback and the Area Chair for their time handling our paper. For convenience, we summarize the key concerns from reviewers and our responses below:

---

**1. Reviewer 8Ccw** requested a detailed energy breakdown of each component, an explanation of our method’s generalizability, and an ablation study of performance for our 1-bit KV configuration in comparison to related works.

**Response: [Done]** We have added a comprehensive energy breakdown analysis clarifying each component’s contribution. We have also provided additional network structures to demonstrate generalizability. Finally, we have conducted an ablation study (with 1-bit KV configuration) that shows our method still achieves state-of-the-art performance.

---



**2. Reviewer CymU** requested several additions, including reporting the energy consumption of the Softmax and LayerNormalization operations, providing inference time results, adding experiments on CNN architectures to increase the generalizability, performing an ablation study on the performance and cost of our QNN-to-SNN conversion, and comparing our approach with the related TTFS method.


**Response: [Done]** We have addressed all of these points in the revision. We have included the energy usage of Softmax and LayerNormalization, confirming that their inclusion does not significantly affect our efficiency results. We have also provided inference time measurements and added experiments with convolutional neural networks to demonstrate generalizability. Additionally, we have included an ablation study on the QNN-to-SNN conversion process and a comparison with TTFSFormer. These results show that our method continues to perform best among the compared approaches.

---

**3. Reviewer Ei9k** asked for a discussion of the hardware scale of our proposed physical device and a comparison with linear-attention SNN approaches.

**Response: [Done]** We have added two new sections to address these issues. One section discusses the hardware scale and potential improvements. The other explains how the linear-attention SNN approach relates to our optical SNNs.


---

**4. Reviewer AM8c** requested two ablation studies (one on the QNN-to-SNN conversion process and another comparing the energy consumption of our method with related works under different settings). Additionally, they asked for references supporting our energy consumption equations, comparisons with existing QNN-to-SNN conversion methods, and information on the size of our device.


**Response: [Done]** We have provided all the requested ablation studies. We also added the relevant references for our energy consumption equations, expanded the comparison with other QNN-to-SNN conversion methods, and reported the size of our device along with a discussion of large-scale deployment. **We are pleased that Reviewer AM8c acknowledged our response and raised their score from 6 to 8.**

---

**5. Reviewer YQwJ**: asked for a detailed energy breakdown and experiments comparing energy consumption with related works.

**Response: [Done]** We included a detailed energy breakdown in the revision. We also added an energy-focused ablation study comparing our consumption with related works.

---

Currently, only Reviewer AM8c has responded to our revisions, raising their score from 6 to 8. We appreciate the Area Chair and Senior Area Chair considering further score adjustments based on our responses. We believe we have fully addressed all concerns by providing the requested experiments and discussion. We sincerely appreciate your additional time and effort!

---

### Meta-Review · Area_Chair_9AhP · 2025-12-21

**Summary:**

This paper presents Otters: a hardware-software co-designs tool that leverages  the nature of signal decay in optoelectronic synapses to execute TTFS computation without expensive digital decay and multiplication operations. It involves all the three aspects of the SNN designs: chip fabrication, designing algorithms (i.e., QNN-to-SNN conversion) and power consumption estimation, demonstrating high accuracy as well as strong performance over GLUE while saving a large amount of energy compared with previous SNN models.

The reviewers described the work as originals, well-motivated and ambitious; they appreciated the idea to turn a device feature into a computational primitive operation. The initial concerns included energy accounting, comparison fairness, generalization beyond BERT, hardware scalability and gain attributions. The authors wrote a detailed response, and enriching the paper with new experiments, ablations, clarifications, discussion items. After revisions, almost all concerns were resolved and some of the reviewers even increased their score.

**Reviewer Concerns:**

Shared Concerns:

1. Energy Decomposition and Fairness of Energy Comparison. Reviewers asked for energy consumption breakdowns, clarity on key terms (e.g., data movement vs. compute) and fair baselines under assumptions (bit-width, batch size/sequence lengths). Resolution: The authors included per-component energy tables, and provided energy table and charts for each component. They also identified data movement (e.g., spike movement) as the key contributors. They tabulated constants, added Softmax and LayerNorm costs, and re-run baselines with the single model using reproduction scripts.

2. Where does the gain come from (Quantization vs. Otters Mechanism)? Reviewers asked whether the energy savings and accuracy trade-offs originate primarily from quantization as opposed to optoelectronic TTFS process. Resolution: The Authors perform ablations of the full-precision → QNN loss and the QNN → Otters loss, and they found that the QNN→Otters loss causes on average only -0.6% accuracy drop while delivering significant energy gains. Equal-bit-width comparisons against QNNs and SNNs were included.

3. Generality Beyond BERT / Transformers. Issues were also raised over exclusive application of the resulting to BERT and Transformers. Resolution: Changes include CNN experiments (vgg/resnet on cifar-10)

4. Scalability and Practicality of the Hardware Quantity. The reviewers illustrated that Equation (6) is scalable hardware-wise. Problems were hardware scale, CMOS integration and large-system practicability. Resolution: the authors added a new section to explain device size, scaling limits, 1-bit weight sharing, and BERT-block device numbers. Paths through device scaling and 3D integration are added by the authors, with challenges acknowledged.

Reviewer-Specific Questions:

1. Reviewer 8Ccw: I look for energy breakdown, generalizability and fair 1-bit KV comparison. Dealt with using new tables, more architectures and paired ablation.

2. Reviewer CymU: Agreed on missing Softmax/LayerNorm energy, latency, and CNN generalization, missing QNN-to-SNN training overhead, missing TTFSFormer comparison Clarity. The authors introduced energy terms, latency analysis, CNN experiments, QNN motivation and in-depth TTFSFormer comparison and optimized formatting.

3. Reviewer Ei9k: Asked about the hardware scale and integrate, linear-attention SNNs such as QKFormer. The authors included discussions on area, reuse and 3D integration and noted the complementarity of Otters to algorithmic design.

4. Reviewer AM8c: Asked about scalability, novelty of conversion with respect to prior QNN-to-SNN, energy fairness and separation of quantization vs. conversion were inquired. Requested with ablations, additional comparisons, equal-bit-width comparison and gap analysis in accuracy. This reviewer increased the final evaluation from 6 to 8.

5. Reviewer YQwJ: Requested for energy origin, dominant terms and baseline methods. The authors breakdowned, differences were clarified, recomputations explained and presentations corrected.

**Reviewer Scores:**

I think the reviewer scores are reasonable. Particularly, AM8c has raised score from 6 to 8.

---

### Decision · Program_Chairs · 2026-01-26

Accept (Poster)